# Leukemia stemness and co-occurring mutations drive resistance to IDH inhibitors in acute myeloid leukemia

Feng Wang[1,9], Kiyomi Morita [2,9], Courtney D. DiNardo [2✉], Ken Furudate[2,3], Tomoyuki Tanaka[2], Yuanqing Yan[4], Keyur P. Patel[5], Kyle J. MacBeth[6], Bin Wu [7], Guowen Liu[7], Mark Frattini[6], Jairo A. Matthews[2], Latasha D. Little[1], Curtis Gumbs[1], Xingzhi Song[1], Jianhua Zhang [1], Erika J. Thompson[8], Tapan M. Kadia[2], Guillermo Garcia-Manero [2], Elias Jabbour[2], Farhad Ravandi[2], Kapil N. Bhalla [2], Marina Konopleva [2], Hagop M. Kantarjian [2], P. Andrew Futreal [1✉] & Koichi Takahashi [1,2✉]

Allosteric inhibitors of mutant IDH1 or IDH2 induce terminal differentiation of the mutant leukemic blasts and provide durable clinical responses in approximately 40% of acute myeloid leukemia (AML) patients with the mutations. However, primary resistance and acquired resistance to the drugs are major clinical issues. To understand the molecular underpinnings of clinical resistance to IDH inhibitors (IDHi), we perform multipronged genomic analyses (DNA sequencing, RNA sequencing and cytosine methylation profiling) in longitudinally collected specimens from 60 IDH1- or IDH2-mutant AML patients treated with the inhibitors. The analysis reveals that leukemia stemness is a major driver of primary resistance to IDHi, whereas selection of mutations in *RUNX1/CEBPA* or *RAS-RTK* pathway genes is the main driver of acquired resistance to IDHi, along with *BCOR*, homologous *IDH* gene, and *TET2*. These data suggest that targeting stemness and certain high-risk co-occurring mutations may overcome resistance to IDHi in AML.

---

[1] Department of Genomic Medicine, The University of Texas MD Anderson Cancer Center, Houston, TX, USA. [2] Department of Leukemia, The University of Texas MD Anderson Cancer Center, Houston, TX, USA. [3] Department of Oral and Maxillofacial Surgery, Hirosaki University Graduate School of Medicine, Hirosaki, Aomori, Japan. [4] Department of Neurosurgery, The University of Texas Health Science Center at Houston, Houstont, TX, USA. [5] Department of Hematopathology, The University of Texas MD Anderson Cancer Center, Houston, TX, USA. [6] Celgene Corporation, Summit, NJ, USA. [7] Agios Pharmaceuticals, Cambridge, MA, USA. [8] Department of Genetics, The University of Texas MD Anderson Cancer Center, Houston, TX, USA. [9]These authors contributed equally: Feng Wang, Kiyomi Morita. ✉email: cdinardo@mdanderson.org; afutreal@mdanderson.org; ktakahashi@mdanderson.org

Somatic mutations in isocitrate dehydrogenase 1 and 2 (*IDH1* and *IDH2*) can be detected in ~20% of patients with acute myeloid leukemia (AML)[1]. Mutations are almost exclusively found in the Arg132 (R132) residue in IDH1 and Arg140 (R140) or Arg172 (R172) residues in IDH2. Wild-type IDH1 and IDH2 catalyze the oxidative decarboxylation of isocitrate to produce α-ketoglutarate (α-KG). On the other hand, mutant IDH1 and IDH2 acquire neomorphic catalytic activity and produce an oncometabolite, (*R*)-2-hydroxyglutarate [(*R*)-2HG or 2HG][2,3], which competitively inhibits α-KG-dependent enzymes, such as the ten-eleven translocation (TET) family of DNA hydroxylases, lysine histone demethylases, and prolyl hydroxylases[4–6]. As a result, IDH-mutant AML exhibits CpG hypermethylated phenotype and increased histone methylation, leading to an aberrant gene expression profile and differentiation arrest[7,8].

Allosteric inhibitors to IDH-mutant proteins (e.g., enasidenib for mutant IDH2 and ivosidenib for mutant IDH1) suppress 2HG production[9], and demonstrate an ~40% overall response rate (ORR) in patients with IDH1- or IDH2-mutant relapsed and refractory AML[10,11]. Clinical responders to the inhibitors show improvement in tri-lineage hematopoiesis and reduction of leukemic blasts. In the majority of the responders, *IDH* mutations are stably detected in matured neutrophils, indicating that the clinical response to the inhibitors is mediated by the terminal differentiation of leukemic blasts[9]. This mechanism of action is consistent with the observations in preclinical models[12,13] and patient-derived xenograft models[14], as well as in longitudinally profiled hematopoietic stem cell populations from patients who responded to enasidenib[15].

While the clinical response to IDHi can be durable, primary and secondary resistance to single-agent therapy are major clinical challenges[10,11]. In a phase 2 study of enasidenib, co-occurrence of *NRAS* mutations or high co-mutation burden were associated with a poor response to the drug[9]. Intlekofer and colleagues reported three cases that developed secondary resistance to enasidenib or ivosidenib[16]. These cases acquired second-site mutations in the IDH2 dimer interface (p.Q316E and p.I391M) or IDH1 p.S280F, which were predicted to interfere with the IDHi binding. The same group of investigators also reported four cases of IDH isoform switching, which refers to the emergence of the mutation in homologous *IDH* gene counterpart during the inhibition of the other IDH mutant (e.g., emergence of *IDH1* mutation during IDH2 inhibition, and vice versa; to avoid confusion, we will call this phenomenon IDH homolog switching in this paper)[17]. In addition, Quek and colleagues studied paired samples at baseline and relapse in 11 AML patients treated with enasidenib[15]. They did not find the second-site mutations, but observed diverse patterns of clonal dynamics (including IDH homolog switching) or selection of subclones associated with the relapse.

While the data from the small case series are accumulating, the entire landscape of clonal heterogeneity and its association with IDHi resistance has not been elucidated. Moreover, the evidence accumulated so far has been restricted to the association between gene mutations and IDHi resistance. To what extent, DNA methylation changes or gene expression profiles are associated with clinical resistance to IDHi is not well understood.

In this work, we perform an integrated genomic analysis combining DNA sequencing, RNA sequencing, and methylation profiling microarray on bone marrow samples collected longitudinally from AML patients treated with IDHi, and describe genetic and epigenetic correlates of response to IDHi. The analysis reveals that gene expression signatures with stemness is associated with primary resistance to IDHi, whereas selection of the resistant mutations plays role in acquired resistance to the drugs. These data add insights into the resistance mechanisms of IDHi in AML.

## Results

**Clinical characteristics of the studied patients**. Clinical characteristics of the 60 patients are provided in the Table 1. Thirty-eight (63%) patients were *IDH2*-mutated, 21 (35%) were *IDH1*-mutated, and 1 (2%) had both mutations. Thirty-eight (63%) patients were treated with enasidenib and 22 (37%) with ivosidenib. The selection of patients was solely based on availability of samples, resulting with a cohort of 30 clinical responders (51%) and 29 nonresponders (49%; response not evaluable in one patient). ORR was not significantly different between patients treated with IDH1 inhibitor (ORR 55%) and IDH2 inhibitor (ORR 49%; $P = 0.821$). Among the 30 responders, 20 patients relapsed after a median duration of response of 4.4 months (interquartile range: 2.7–8.0). Compared with the samples that were not analyzed in this study (due to the lack of sample availability), the studied cohort were older and contained more enasidenib-treated cases (Table S1).

**Co-occurring *RUNX1* or *RAS* signaling mutations are associated with primary resistance to IDH inhibitors**. Targeted deep sequencing of pretreatment samples identified 262 high-confidence somatic mutations (177 single-nucleotide variants [SNVs] and 85 small insertions and deletions [indels]) in 36 cancer genes (Fig. 1A). Mutations that co-occurred with *IDH1/2* mutations were most frequently found in *DNMT3A* ($N = 26$,

**Table 1 Clinical characteristics of the 60 patients treated with IDH inhibitors.**

| Characteristics | Median | IQR |
|---|---|---|
| WBC | 1.8 | 1.1–3.1 |
| ANC | 0.2 | 0.0–0.6 |
| HGB | 9.4 | 8.6–10.4 |
| PLT | 47 | 24–76 |
| BM blast, % | 41 | 20–63 |
| PB blast, % | 6 | 0–28 |
| Age | 72 | 60–77 |
| | No. | % |
| Diagnosis | | |
| AML | 55 | 92 |
| MDS | 4 | 7 |
| CMML | 1 | 2 |
| Karyotype | | |
| Normal | 24 | 40 |
| Complex | 13 | 22 |
| Trisomy 8 | 10 | 17 |
| Deletion 5 | 3 | 5 |
| 11q23 | 3 | 5 |
| Other | 13 | 22 |
| Sex | | |
| Female | 25 | 42 |
| Male | 35 | 58 |
| Treatment | | |
| Ivosidenib | 22 | 37 |
| Enasidenib | 38 | 63 |
| Best response | | |
| CR | 12 | 20 |
| CRp | 7 | 12 |
| MLFS | 7 | 12 |
| HI with platelet and neutrophil response | 1 | 2 |
| PR | 3 | 5 |
| SD | 28 | 47 |
| PD | 1 | 2 |
| Not assessed | 1 | 2 |
| IDH-differentiation syndrome | | |
| Yes | 11 | 18 |
| No | 49 | 82 |

Source data are provided as a Source data file.
*IQR* interquartile range, *WBC* white blood cells, *ANC* absolute neutrophil count, *HGB* hemoglobin, *PLT* platelets, *BM* bone marrow, *PB* peripheral blood, *No.* number, *AML* acute myeloid leukemia, *MDS* myelodysplastic syndrome, *CMML* chronic myelomonocytic leukemia, *CR* complete remission, *CRp* CR with incomplete platelet recovery, *MLFS* morphological leukemia-free state, *HI* hematological improvement, *PR* partial remission, *SD* stable disease, *PD* progressive disease.

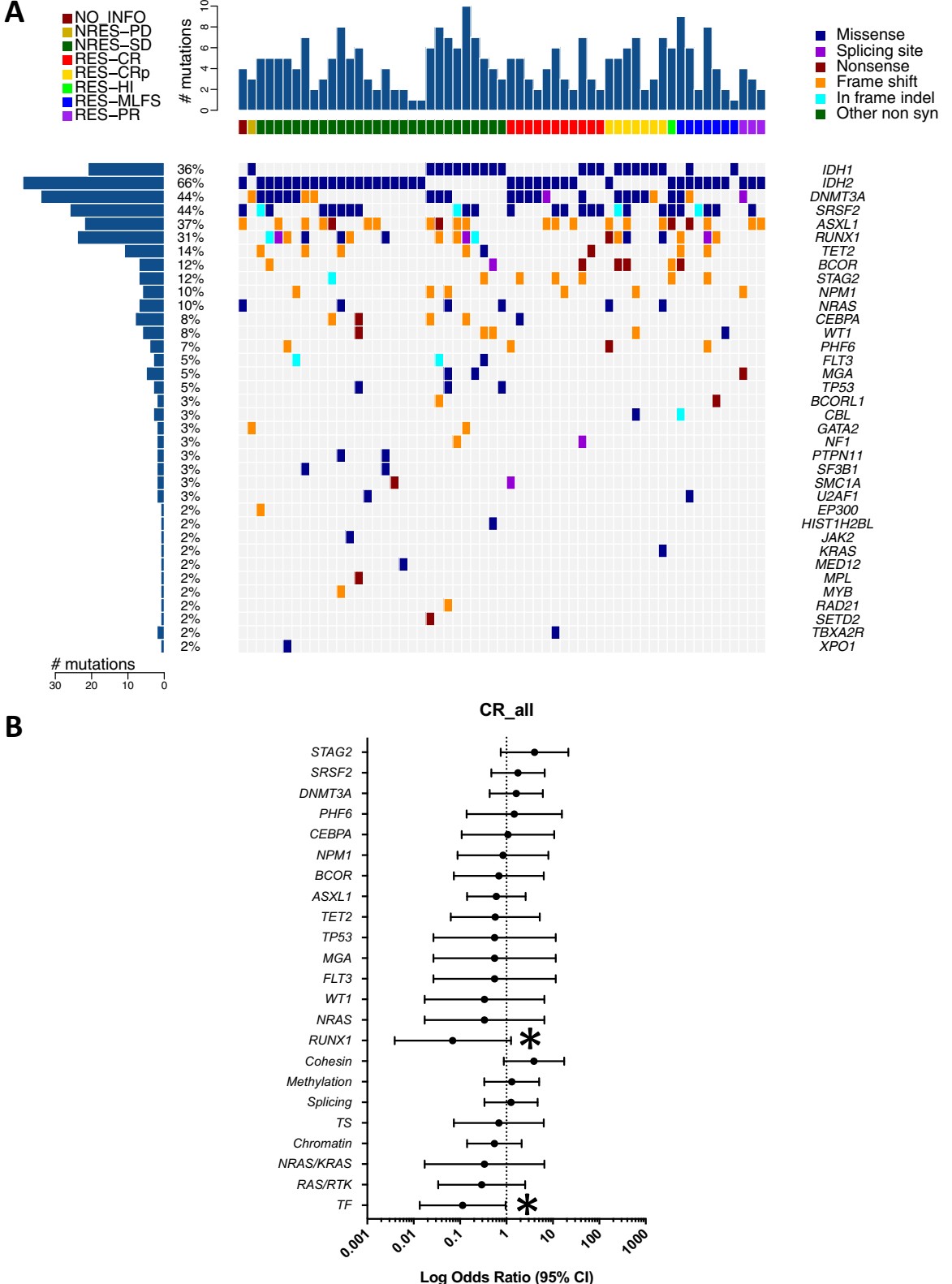

**Fig. 1 Mutational landscape of IDHi-treated AML patients and their association with clinical response. A** Landscape of high-confidence somatic mutations detected in baseline samples by sequencing with a 295-gene panel. Legend for the best response is located at the top left, while legend for the mutation classification is located at the top right. Baseline mutation data are available for 59 patients. **B** Forrest plot showing enrichment of the mutations at baseline against complete remission (CR) by logarithmic odds ratio. Two-sided Fisher's exact test was performed. *$P < 0.05$ ($P = 0.012$ for *RUNX1*; $P = 0.037$ for TF). Circles (center of the error bars) represent odds ratios. The error bars represent 95% confidence interval of odds ratio. Baseline mutation data are available for 59 patients, out of which 11 achieved CR. Genes mutated in three or more patients are plotted. TS tumor suppressor, TF transcription factors. Source data are provided as Source data files.

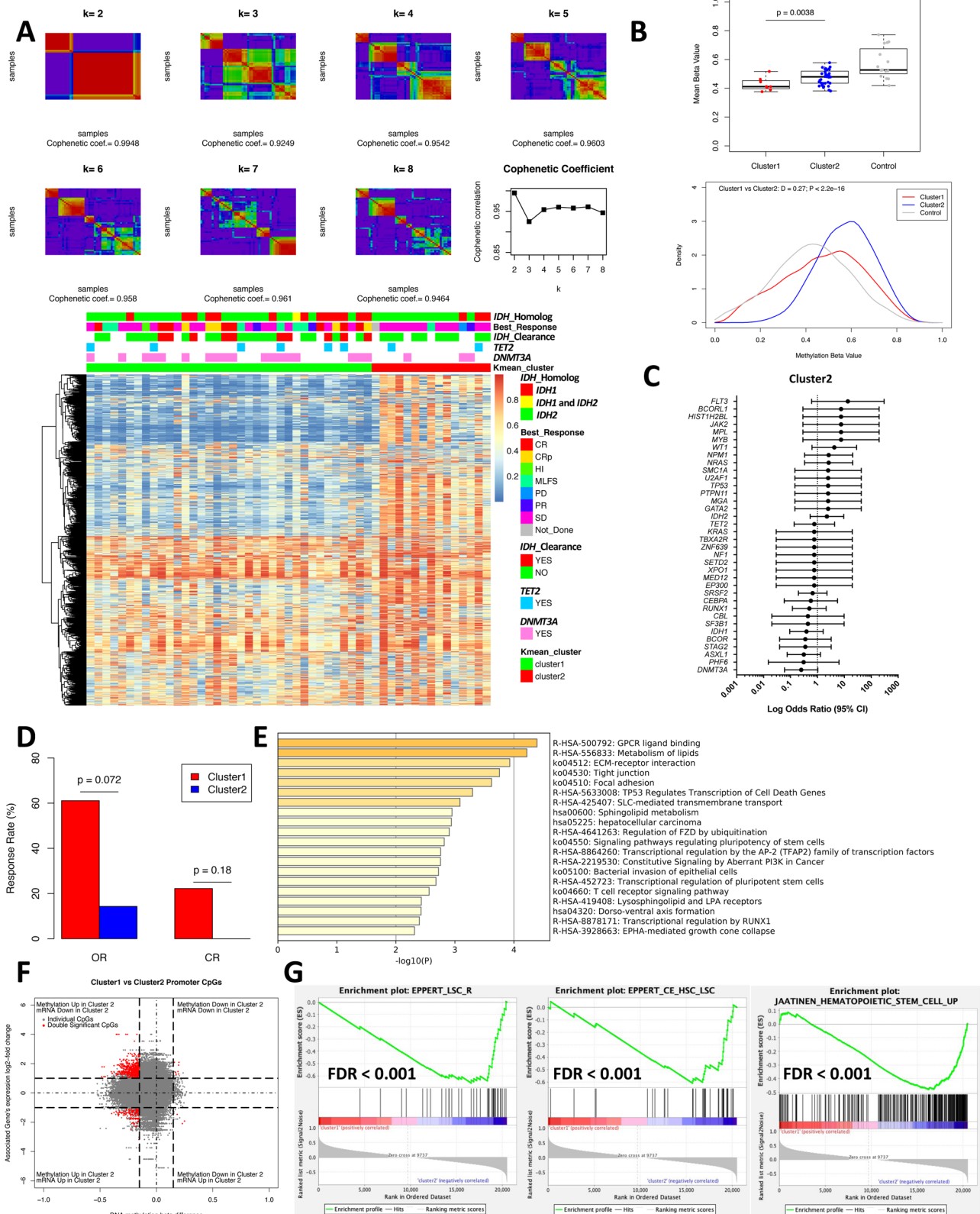

44%), *SRSF2* (*N* = 26, 44%), *ASXL1* (*N* = 22, 37%), and *RUNX1* (*N* = 18, 31%). Relative timing of the mutation accrual was inferred based on the estimated cancer cell fraction of the co-occurring mutations. In relative to *IDH1/2* mutations, mutations in *SRSF2*, *U2AF1*, *DNMT3A*, and *RUNX1* were predicted to have occurred earlier, whereas mutations in oncogenic *RAS* pathway genes (*NF1*, *PTPN11*, *CBL*, and *NRAS*) were likely acquired later

(i.e., subclonal, Fig. S1). Mutations associated with clonal hematopoiesis (e.g., *DNMT3A*, *ASXL1*, *TET2*, *SRSF2*, and others) were common, reflecting high median age of this cohort (median 72 years, Table 1).

The analysis of co-occurring mutations and clinical response revealed that patients with concurrent *RUNX1* mutations had significantly inferior complete remission (CR) rate (*P* = 0.012,

**Fig. 2 Analysis of DNA methylation at baseline samples reveals two distinct clusters associated with treatment response. A** Consensus $k$-mean clustering of promoter methylation data at baseline revealed two distinct clusters. Methylation data are based on methylation beta value. Promoter CpG probes from top 1% most variably methylated CpG probes were selected for the analysis. Responders were defined as patients achieved best response of CR, CRp, MLFS, PR, and HI. Nonresponders were defined as patients achieved best response of PD and SD. **B** Top: Box plot comparing mean methylation beta value of top 1% most variably methylated CpGs among baseline samples for cluster 1 ($N = 36$) and cluster 2 ($N = 15$). *IDH1/2* wild-type AML samples ($N = 8$) are used as control. Box plot shows the minimum, first quartile (Q1), median, third quartile (Q3), and maximum. Two-sided Student's $t$ test was performed. Bottom: Density distribution of top 1% most variably methylated CpG probes with methylation beta values comparing baseline samples of cluster 1 and cluster 2. Two-sided Kolmogorov–Smirnov test was performed. *IDH1/2* wild-type AML samples ($N = 8$) are used as control. **C** Forrest plot showing enrichment of the mutations at baseline against cluster 2 by logarithmic odds ratio. Circles (center of the error bars) represent odds ratios. The error bars represent 95% confidence interval of odds ratio. Baseline mutation data with clustering information are available for 51 patients, out of which 15 is in cluster 2. **D** Bar plot comparing the overall response (OR) and CR rate between cluster 1 ($N = 36$) and cluster 2 ($N = 14$) patients. Two-sided Fisher's exact test was performed. **E** Metascape analysis of hypermethylated promoter DMPs. **F** Starburst plot showing integrated analysis of gene expression and promoter methylation changes between cluster 1 and cluster 2. Of 215,521 promoter CpGs, 704 showed statistically significant differential methylation and differential expression between cluster 1 and cluster 2. A total of 558 of 704 double significant CpGs showed promoter hypermethylation and downregulation of the gene expression in cluster 2. **G** Gene set enrichment analysis (GSEA) comparing gene expression profiles between the two clusters revealed upregulation of genes associated with leukemia stem cells (LSCs) in cluster 2. Source data are provided as Source data files.

Fig. 1B and Fig. S2). None of the patients with concurrent *NRAS* mutations ($N = 6$), previously associated with a poor CR rate with enasidenib[9], achieved CR (Fig. 1B and Fig. S2). Of note, none of the patients with co-occurring *TP53* ($N = 3$) or *FLT3* mutations ($N = 3$) responded to the therapy, while the association was not statistically significant due to the small number of cases. When genes were grouped with functional pathways, co-occurring mutations in hematopoietic differentiation transcription factor (TF) genes (*RUNX1*, *CEBPA*, and *GATA2*) were associated with a significantly worse CR rate ($P = 0.037$), whereas co-occurring mutations in cohesin genes (*STAG2*, *SMC1A*, and *RAD21*) were associated with a trend toward better response ($P = 0.083$, Fig. 1B). In the current cohort, we did not find a significant association between treatment response and the total number of co-occurring mutations (Fig. S3).

**Leukemia stemness is associated with primary resistance to IDHi.** Consensus $k$-means clustering of promoter methylation profiles in pretreatment samples revealed two major clusters: cluster 1 with relative hypomethylation and cluster 2 with relative hypermethylation (Fig. 2A, B and Fig. S4). *DNMT3A* mutations were more frequent in cluster 1 compared to cluster 2 (Fig. 2C), which likely accounts for the relative hypomethylation of the cluster, since co-occurrence of *DNMT3A* mutations with *IDH* mutations has been shown to cause methylation antagonism[18]. None of the other driver mutations had a significant correlation with the methylation-based clusters (Fig. 2C).

Notably, cluster 2 (hypermethylated cluster) was associated with poor response to IDHi (Fig. 2D). The analysis of differentially methylated probes (DMP) between the two clusters showed that promoters in genes related to hematopoietic differentiation, such as RUNX1 transcriptional regulation and transcriptional regulation of stem cells, were significantly hypermethylated in cluster 2 (Fig. 2E). Integrated analysis of promoter methylation and gene expression found that majority of the hypermethylated DMPs led to downregulation of gene expression (Fig. 2F and Supplementary Data 1). Among those genes included megakaryocytic differentiation genes (Fig. S5), suggesting that hypermethylated phenotype of cluster 2 is associated with downregulation of differentiation genes.

We then analyzed the difference in gene expression profiles between the two clusters. Gene set enrichment analysis (GSEA) comparing gene expression profiles between the two clusters revealed upregulation of genes associated with leukemia stem cells (LSCs) in cluster 2 (Fig. 2G). To further explore the molecular drivers of cluster 2 phenotype, we performed NetBID analysis, a data-driven network-based Bayesian inference that identifies

hidden drivers in a given transcriptome[19]. Among the top driver TFs genes enriched in cluster 2 included *FOXC1*, which is one of the critical regulators of LSC function (Fig. 3A)[20]. Additional driver identified for cluster 2 included *CD99*, which encodes essential signaling proteins for LSC (Fig. 3B)[21]. In addition, *DNMT3A* was identified as one of the drivers in cluster 2, that is consistent with the relative absence of *DNMT3A* mutations in the cluster because *DNMT3A* mutations are generally loss-of-function mutations (Fig. 3B). Together, these data suggest that cluster 2 is enriched with samples manifesting increased stemness, which might be associated with resistance to IDHi.

To determine the association between leukemia stemness and IDHi resistance, we calculated the 17-gene LSC score (LSC17) for each sample, which has been associated with leukemia stemness and chemoresistance in AML[22]. Nonresponders to IDHi had a significantly higher LSC17 compared with responders (Fig. 3C). LSC17 predicted response to IDHi (CR) with AUROC (area under the curve of receiver operating curve) of 0.785 ($P = 0.028$), which was better than the predictability of ELN cytogenetic risks (AUROC = 0.533), *RUNX1* mutations (AUROC = 0.522), or *RAS-RTK* mutations (AUROC = 0.6; Fig. 3D, E). Multi-logistic regression analysis also showed that LSC17 was the significant covariate predicting response to IDHi (Table 2). Collectively, these data indicate increased stemness as one of the mechanisms of primary resistance, and the stemness score as a potential predictive biomarker for IDHi response.

**DNA methylation changes after IDHi.** We then analyzed the changes in DNA methylation after IDHi therapy. While there were some heterogeneities among samples, overall, we observed significant demethylation after IDHi (Fig. 4A). Demethylation was observed in samples regardless of the methylation-based clusters (cluster 1 vs. 2) or treatment response. Consistent with this, plasma 2HG was also suppressed after IDHi in most of the patients regardless of the clusters and treatment responses (Fig. 4B). While cluster 1 and cluster 2 both exhibited incremental demethylation after the therapy, cluster 2 remained relatively hypermethylated after the therapy compared to cluster 1 (Fig. 4A). The analysis of methylation changes in individual CpGs revealed that the same set of CpGs were demethylated between cluster 1 and cluster 2 (Fig. 4C). Consistent with this, posttreatment DMPs between the two clusters were largely the same with those at baseline, with most of the DMPs remained hypermethylated in cluster 2 (Fig. 4D, E). The same trend was observed when we compared the methylation changes in individual CpGs between responders and nonresponders (Fig. 4F). GSEA comparing gene expression profiles of posttreatment samples in

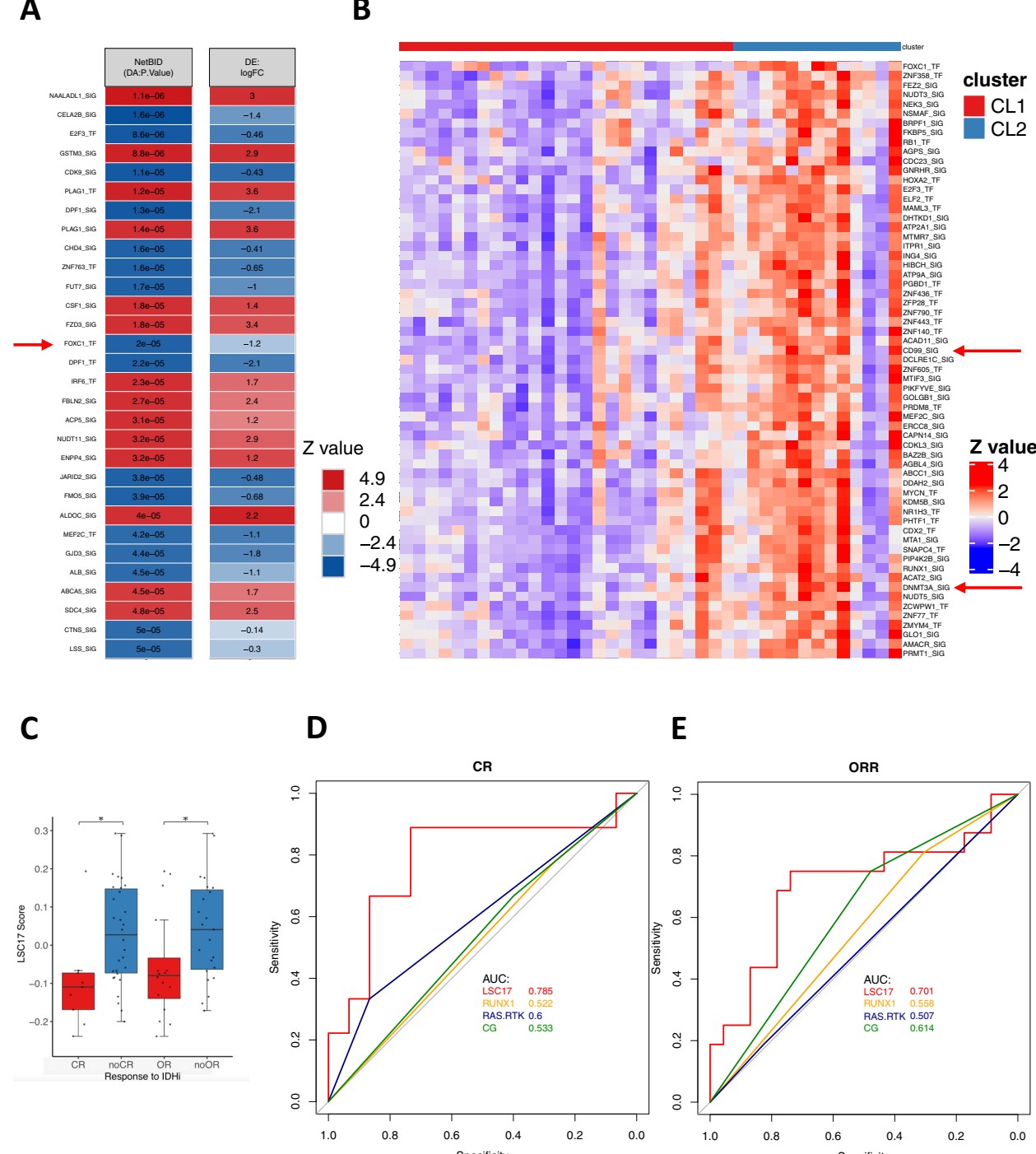

**Fig. 3 Leukemia stemness is associated with primary resistance to IDHi. A** List of top driver transcription factors (TF) and signaling genes (SIG) identified by NetBID2 analysis by comparing gene expression profiles between cluster 1 and cluster 2 (*P* < 0.01 was used as the significance cutoff). Drivers identified for cluster 1 and cluster 2 are colored with red and blue, respectively. **B** Heatmap of NetBID-based activity of top drivers in cluster 2. Samples in cluster 1 (CL1) are labeled as red, while cluster 2 (CL2) are labeled as blue. **C** LSC17 score was calculated for each baseline sample and compared between patients achieving CR (*N* = 9) vs. not (*N* = 30) and OR (*N* = 16) vs. not (*N* = 23). *P* < 0.05 (*P* = 0.011 for CR vs. non-CR; *P* = 0.037 for OR vs. non-OR). Box plot shows the minimum, first quartile (Q1), median, third quartile (Q3), and maximum. **D, E** Receiver operating curve (ROC) for predicting CR or OR with LSC17 score, *RUNX1* mutation status, *RAS-RTK* mutation status, and ELN cytogenetic risk classification. Source data are provided as Source data files.

cluster 1 and 2 showed that LSC-associated genes are still upregulated in cluster 2 (Fig. 4G), suggesting that the stemness is not reversed by IDHi. Collectively, these results suggest that incremental changes in DNA methylation are likely the consequence of 2HG suppression by IDHi therapy and does not necessarily contribute to the clinical response.

**Clonal selection of driver mutations frequently accompanies relapse after IDHi.** We then investigated the mechanisms of acquired resistance to IDHi by analyzing mutational changes in longitudinal samples collected after IDHi therapy. With regards to the *IDH* mutations, variant allele frequency (VAF) of the mutations stayed unchanged in 71% of the responders (*N* = 17),

**Table 2 Multi-logistic regression analysis against CR by considering following variables: LSC17 score (as a continuous variable), *RUNX1* mutation status (mutated vs. wild type), *RAS-RTK* mutation status (mutated vs. wild type), and ELN high-risk cytogenetics classification (high risk vs. others).**

|  | Estimate | Std. error | z value | P value |
|---|---|---|---|---|
| LSC17_score | −9.71868 | 4.307062 | −2.25645 | 0.024042 |
| *RUNX1* mutation | −0.31926 | 1.016707 | −0.31402 | 0.753508 |
| *RAS/RTK* mutation | 1.64524 | 1.086779 | 1.513868 | 0.130059 |
| ELN high-risk cytogenetics | −0.30335 | 0.945247 | −0.32092 | 0.748271 |

whereas 29% of the responders ($N = 7$) had a substantial reduction (>=75% decrease of VAF) or clearance of the mutations at response (Fig. S6). The baseline VAF or the types of *IDH* mutations did not predict the clearance of the mutations (Fig. S7). Also, there was no correlation between *IDH* mutation clearance and the patterns of co-occurring mutations (Fig. S8). Consistent with the previous report, patients with *IDH* mutation clearance had a better relapse-free survival compared to those without clearance (Fig. S9)[10]. In nonresponders, *IDH* VAF were mostly unchanged, but three patients had a substantial reduction on therapy (Fig. S10).

Co-occurring mutations demonstrated variable dynamics during therapy. Emergence of previously undetectable mutations or selection of subclonal mutations frequently accompanied the relapse or disease progression (Fig. 5A). Among the 18 patients with pretreatment and relapse pairs, emerging or selected mutations were detected in 16 (89%) patients at the time of relapse (Fig. 5A). Mutations that were frequently acquired or selected at relapse involved *BCOR* in four cases, followed by *RUNX1*, *KRAS*, and *NRAS* in three cases each (Fig. 5B). IDH dimer-interface mutations were not detected in this cohort. IDH homolog switching occurred in one case. Overall, relapse-associated mutations involved *RAS-RTK* pathway (39%), chromatin structure (28%), hematopoietic TFs (28%), and DNA methylation pathways (17%; Fig. 5B). These relapse-associated mutations were remarkably similar to those associated with poor initial response to IDHi (Fig. 1A, B), highlighting their crucial role in the clinical resistance of IDHi. To determine whether the relapse-associated mutations are part of or independent of *IDH* mutations, we performed a single-cell DNA sequencing (scDNA-seq) in one relapsed sample. In UPN2394529, the relapse was associated with emerging *KRAS* p.Q61H and *NRAS* p.G12S mutations (Fig. 5C). The scDNA-seq revealed that the emerging *NRAS* and *KRAS* mutations were independent of *IDH2* mutation, indicating that non-IDH-mutant clones were driving the relapse in this case (Fig. 5D).

GSEA comparing the gene expression profiles between pretreatment and relapse pairs showed the enrichment of genes downregulated in LSC in relapsed samples (Fig. S11), which contrasts to the samples with primary resistance, indicating for a difference between primary resistance and acquired resistance. Instead, relapsed samples were associated with upregulation of genes in E2F targets, TNF alpha signaling via NF-kappa B, and G2M checkpoint genes, that are consistent with the frequent acquisition of *RAS-RTK* pathway mutations at relapse (Fig. 5E). Collectively, these results underscore the role of co-occurring mutations, particularly *RUNX1* and *RAS-RTK* pathway mutations in acquired resistance to IDHi, and that the co-occurring mutations can be part of or independent of IDH-mutant clones.

**Mapping genetic and epigenetic evolution during IDHi therapy in individual cases.** The heterogeneity in genetic evolution and methylation changes after IDHi prompted us to investigate the dynamic changes in genome and epigenome, during IDHi therapy at individual patient level. Sixteen patients (5 responders and 11 nonresponders) had a set of multidimensional data available at longitudinal time points to map the evolution of somatic mutations and DNA methylation along with clinical parameters and plasma 2HG in individual cases. This analysis identified three major patterns of epigenetic evolution in responders, which correlated with the underlying genetic evolution.

In the first pattern, IDHi effectively suppressed plasma 2HG and bone marrow DNA methylation level at response and the suppression of both markers continued at relapse. The pattern was observed in two cases, UPN1825001 and UPN2463247, of which the relapse was associated with growing *KRAS* mutations. In both cases, plasma 2HG remained suppressed at the time of disease progression, which correlated with sustained suppression of DNA methylation (Fig. 6A, B).

In the second pattern, IDHi similarly suppressed both plasma 2HG and bone marrow DNA methylation at response; however, at relapse, we observed de-suppression of DNA methylation, while plasma 2HG remained low. This pattern was associated with emerging *TET2* mutations at relapse, which is consistent with the role of *TET2* mutation in causing hypermethylation phenotype (observed in UPN2297625 and UPN2620771; Fig. 6C, D). In both pattern 1 and 2, IDHi remained functionally active at relapse (i.e., ongoing 2HG suppression).

The third pattern was observed in one case UPN2370759, which was consistent with the IDH homolog switching previously described[17]. The case initially had *IDH2* mutation and was treated with AG221; however, the relapse was associated with an emergence of *IDH1* p.R132C mutation. In this case, both plasma 2HG and methylation levels increased at relapse, which is consistent with the emergence of *IDH1* mutation during IDH2 inhibition (Fig. 6E). The *IDH1* p.R132C mutation that emerged was not detectable at baseline by both targeted sequencing and the digital droplet PCR assay (sensitivity 0.01%), making it more likely that the mutation was acquired de novo at relapse (Fig. S12).

In nonresponders, 9 out of 11 samples showed co-suppression of plasma 2HG and DNA methylation after IDHi therapy, while it did not lead to clinical response in these patients. In two cases, despite suppression of plasma 2HG, we did not observe demethylation. Underlying mechanisms of this discrepancy is not clear. Genetic and epigenetic evolution of nonresponders are shown Fig. S13.

## Discussion

Using a multipronged genomic analysis on longitudinally collected samples from the clinical trials, we studied genetic and epigenetic correlates of response to IDHi in AML. While confirming previous findings about the role of certain co-occurring mutations (*RAS* and *RUNX1*) in primary resistance to IDHi[9], we additionally revealed that leukemia stemness is associated with IDHi primary resistance. In the current cohort, higher LSC17 score was the strongest predictor of response to IDHi. Since the clinical activity of IDHi is driven by the induction of terminal differentiation of leukemic blast[14], it is plausible that stemness phenotype causes inherent resistance to the differentiating mechanism of action of IDHi monotherapies. The underlying mechanisms driving stemness in IDH-mutant AML is unclear. Increased stemness was associated with hypermethylated phenotype (cluster 2) in this cohort, which had a further

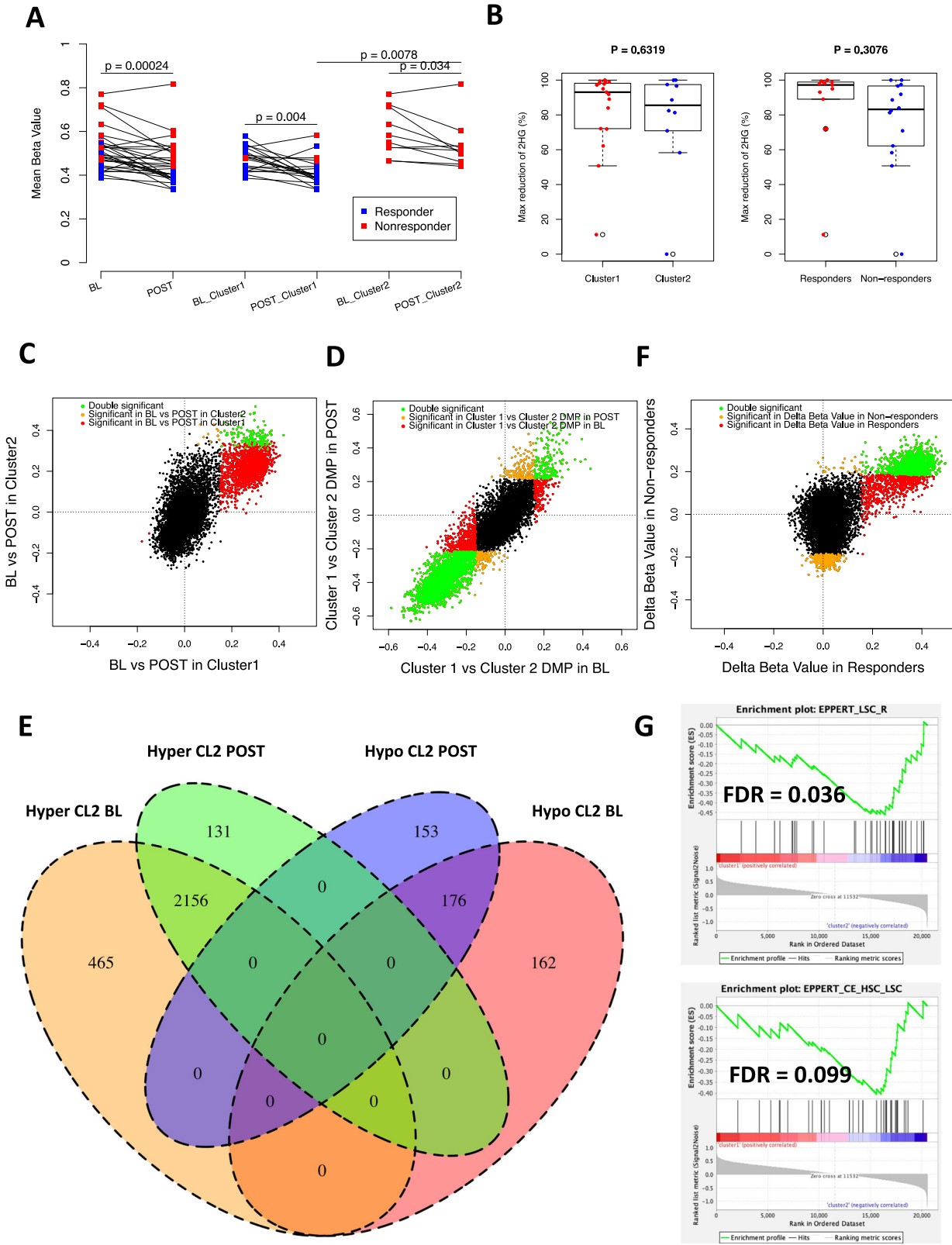

association with the absence of co-occurring *DNMT3A* mutations. While we found correlation between hypermethylated DMPs and downregulation of genes associated with hematopoietic differentiation, whether the hypermethylated phenotype (or the lack of co-occurring *DNMT3A* mutations) is directly causing an increased stemness is not clear. Since *IDH* mutations broadly affects methylation status, including enhancers and histones, further investigation is needed to understand the mechanisms driving the stemness in *IDH*-mutated AML, and the connection between the stemness and hypermethylation status.

**Fig. 4 DNA methylation changes after IDHi. A** Longitudinal trend of methylation level at baseline (BL) and posttreatment (POST) for all, cluster 1 and cluster 2 patients. Responders and nonresponders are color coded. Two-sided Student's $t$ test was performed. **B** Box plot showing maximum reduction of plasma 2HG levels after IDHi treatment (%) in cluster 1 ($N = 17$) and cluster 2 ($N = 10$) patients, as well as responders ($N = 13$) and nonresponders ($N = 14$). Box plot shows the minimum, first quartile (Q1), median, third quartile (Q3), and maximum. Two-sided Student's $t$ test was performed. **C** Scatterplot showing correlation of the longitudinal methylation changes between cluster 1 and cluster 2 patients. Each dot represents a CpG probe and was colored based on its significance in the longitudinal differentially methylation test in either cluster 1 or cluster 2 patients. The $X$-axis represents the differential methylation level between BL and POST samples (i.e., beta value at BL minus beta value at POST) in cluster 1 patients and the $Y$-axis represents the differential methylation level between BL and POST samples in cluster 2 patients. **D** Scatterplot showing correlation of the intercluster methylation differences between BL and POST time points. Each dot represents a CpG probe and was colored based on its significance in the intercluster differentially methylation test at either BL or POST time points. The $X$-axis represents the differential methylation level between cluster 1 and cluster 2 in BL samples (i.e., beta value of cluster 1 minus cluster 2) and the $Y$-axis represents the differential methylation level between cluster 1 and cluster 2 in POST samples. **E** Venn diagram showing the overlapped DMPs between cluster 1 and cluster 2 at baseline (BL) and posttreatment (POST). Among 2621 DMPs hypermethylated in cluster 2, 2156 overlapped between BL and POST, suggesting that most of the hypermethylated DMPs in cluster 2 were the same before and after treatment. **F** Scatterplot showing correlation of the longitudinal methylation changes between responders and nonresponders. Each dot represents a CpG probe and was colored based on its significance in the longitudinal differentially methylation test in either responders or nonresponders. The $X$-axis represents the differential methylation level between baseline and response samples in responders (i.e., beta value at BL minus beta value at POST) and the $Y$-axis represents the differential methylation level between baseline and non-response samples in nonresponders. **G** GSEA analysis comparing gene expression of posttreatment samples between cluster 1 and cluster 2 showed that LSC genes are still upregulated in cluster 2 posttreatment. Source data are provided as Source data files.

Nonetheless, our findings, while require validation in independent cohort, offer a possibility that stemness signatures may function as a predictive biomarker for IDHi response.

Co-occurring mutations and the selection of resistant mutations were also critical factors for IDHi resistance, particularly in the setting of acquired resistance[15]. There was no single dominant gene mutation associated with the resistance, however, the mutations implicated for the resistance were consistent in both primary and acquired resistance settings, underscoring their role in clinical resistance to IDHi. One of the major pathways affected by the mutations were hematopoietic differentiation TFs, particularly involving *RUNX1*. *RUNX1* co-mutation(s) at baseline was associated with lower CR rate in our cohort. *RUNX1* mutations were also among the most frequently acquired or selected mutations at relapse. In addition, four out of five patients with co-occurring *CEBPA* mutations did not respond to IDHi and the mutation were also acquired at relapse in two patients (Fig. 5A). Since *RUNX1* and *CEBPA* both encode essential TFs for hematopoietic and myeloid differentiation, mutations in these genes likely abrogates differentiation signals induced by IDHi, thus contributing to the clinical resistance.

Mutations in the *RAS-RTK* pathway represent another major mechanism of the resistance. The association between co-occurring *RAS* pathway mutations and primary resistance to enasidenib or ivosidenib has been previously reported[9,23]. In our cohort, co-occurrence of *NRAS* mutations at baseline trended with a poor response to the IDHi therapy. In addition, *NRAS* or *KRAS* mutations had been acquired at relapse in nearly 30% of the cases. Intriguingly, in at least one case that had acquired *KRAS* and *NRAS* mutations at relapse (UPN2394529), the mutations did not co-occur with *IDH* mutation by the single-cell sequencing, suggesting that selection of non-IDH clone can also drive relapse.

While co-occurring mutations in *RUNX1/CEBPA* or *RAS-RTK* genes were the major pathways to IDHi resistance, we also observed other less frequent, but intriguing mechanisms. One was an acquired mutation in the homologous gene. The same IDH homolog switching phenomenon was previously reported in two cases of AML treated with ivosidenib[17]. This pattern was associated with an increase in plasma 2HG and DNA hypermethylation at relapse. We also observed acquisition of *TET2* mutation as a likely IDHi resistance mechanism. In contrast with the homolog switching, these *TET2*-acquired cases showed continued suppression of plasma 2HG at relapse, while DNA

hypomethylation did not occur. We also observed frequent acquisition of loss-of-function mutations in the *BCOR* gene at relapse. *BCOR* is part of noncanonical PRC1.1 complex, which acts as a transcription corepressor[24]. It is not yet clear how loss of BCOR function contributes to IDHi resistance, but the data offer hypothesis that BCOR target genes may be involved in IDHi resistance. In our cohort, we did not observe the acquisition of second-site mutations at the dimer interface of IDH1/2 (ref. [16]). This was also not found in 11 cases of post enasidenib relapse analyzed by Quek et al.[15]. A recent study by Choe et al. identified the second-site mutations in 14% of IDH1-mutant AML patients who relapsed after ivosidenib[23]. Although the dimer interface mutations in *IDH1/2* represent a compelling mechanism of acquired resistance to IDHi, further studies are needed to understand the true prevalence of this mechanism. We have reviewed and summarized the available evidence related to the molecular mechanisms of IDHi resistance in Table 3.

This study also analyzed dynamic changes in CpG methylation during IDHi therapy. The drug induced hypomethylation in bone marrow samples that is consistent with the suppression of 2HG (likely through the restoration of TET family protein activity). However, the incremental changes in DNA methylation occurred in the same CpGs among responders and nonresponders, and there was no concordance between DNA methylation changes and clinical response. These data suggest that incremental changes in DNA methylation mirror the 2HG dynamics except in rare cases with co-occurring *TET2* mutations, and do not correlate with clinical response to IDHi.

There are several limitations in our study. First, we could not independently validate the association between stemness signature and IDHi response. This finding needs to be confirmed in an independent cohort of AML patients treated with IDHi. Second, the sample size of this study was underpowered to capture rare molecular predictors of IDHi response, for example, *FLT3* and *TP53*. Results from the several independent studies correlating gene mutations and IDHi response are now available, and meta-analysis of the combined dataset might reveal the entire landscape of gene mutations and their impact on IDHi response. Third, due to the limited amount of the available specimens, multi-omics analyses were not possible in all sample, leading to inconsistencies in data generation among samples (Fig. S14). Fourth, our cohort included heterogeneous patient populations who were at the different stages of their disease and also included small number of patients with MDS/CMML. Also, different types

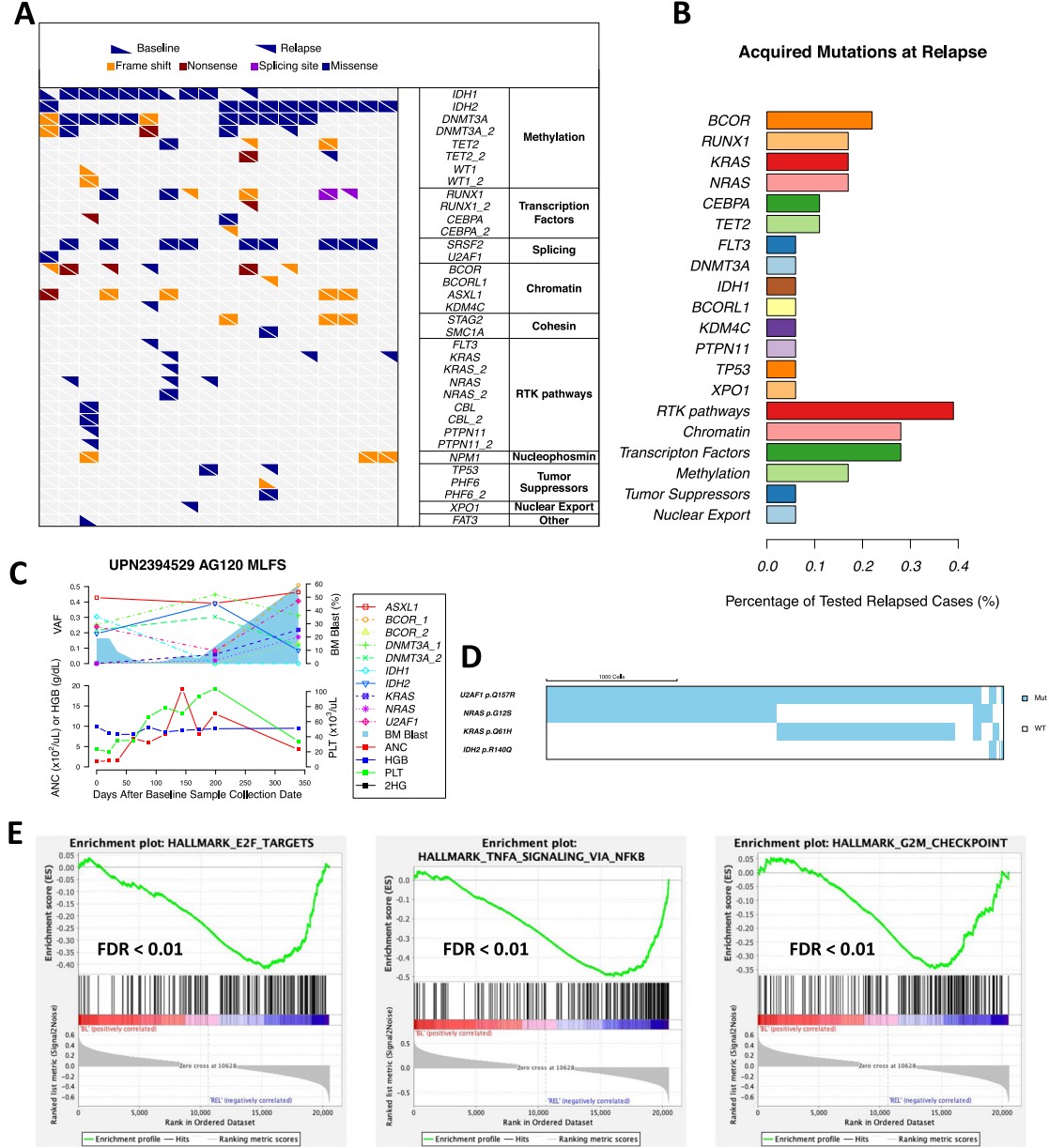

**Fig. 5 Selection of resistant mutations accompanies relapse after IDHi. A** Longitudinal mutation landscape plot showing mutation acquisitions in 16 out of the 18 tested relapsed cases. Each column represents an individual case with differentially shaped triangles representing mutation status in either baseline or relapse. **B** Bar plot showing percentage of tested relapsed cases with acquired mutations in various genes and pathways. **C** The longitudinal trajectory of mutation VAFs, bone marrow (BM) blast counts, absolute neutrophil count (ANC), hemoglobin (HGB) counts, and platelet (PLT) counts in UPN2394529 (**C**). Line plots show mutation VAFs and ANC/HGB/PLT counts. Blue shades represent BM blast counts. **D** Single-cell landscape of selected mutations in UPN2394529. Each column represents one individual cell. A total of 1000 cells scale bar is shown on the top left. **E** GSEA comparing gene expression data from RNA sequencing between baseline and relapse samples showing significant enrichment of E2F targets, TNF alpha signaling via NF-kappa B, and G2M checkpoint genes, in relapse samples. Source data are provided as Source data files.

of IDH inhibitors were given to the patients (enasidenib and ivosidenib). While we do not believe these heterogeneities affect overall conclusion of our study, our findings need to be validated in patients with more uniform characteristics. Lastly, the targeted DNA sequencing might have missed low VAF mutations for mutation clearance and clonal dynamics analysis. With all these limitations in mind, we believe that our study adds insights into genetic and epigenetic mechanisms of resistance to IDHi.

In summary, the molecular profiling of IDHi-treated AML samples revealed that leukemia stemness plays major role in primary resistance to the drug, whereas co-occurring mutations, particularly in hematopoietic TF genes (*RUNX1* and *CEBPA*) and

*RAS-RTK* genes, are critical factors for acquired resistance. These results suggest that combination strategies targeting stemness and co-occurring mutations may improve the therapeutic efficacy of IDHi in AML[25]. The results from ongoing combination therapy trials (IDHi with azacitidine, cytarabine + daunorubicin, MEK inhibitor, or venetoclax) are warranted to understand how these approaches can overcome these resistance mechanisms.

## Methods

**Patients and samples**. We studied 60 patients with relapsed or refractory myeloid malignancies (AML *N* = 55, MDS *N* = 4, and CMML *N* = 1) who received IDH inhibitor therapy in one of the two clinical trials conducted in our institution:

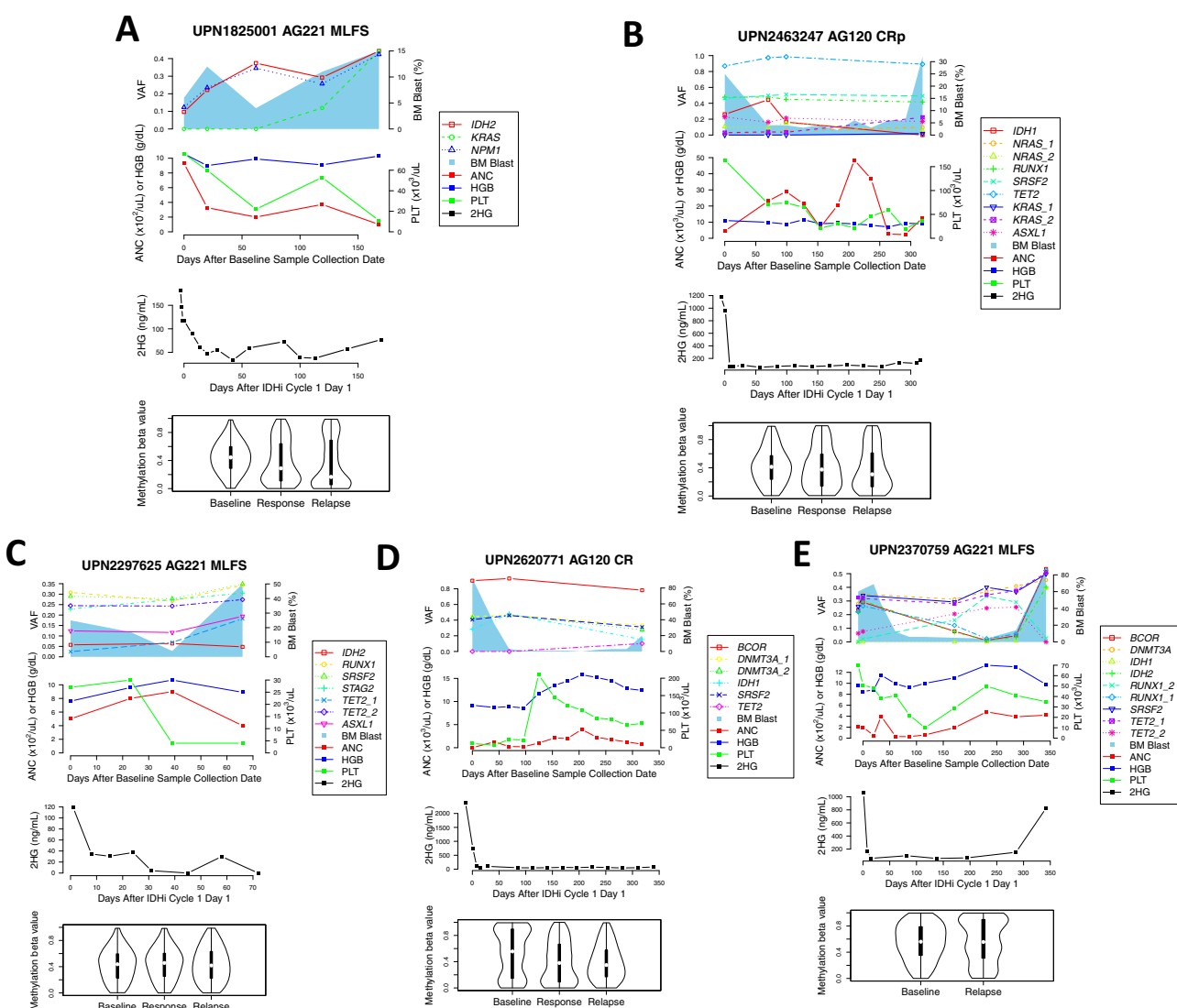

**Fig. 6 Heterogeneous patterns of genetic and epigenetic evolution in AML patients treated with IDHi. A–E** Multidimensional longitudinal plot of mutation VAFs, bone marrow (BM) blast counts, absolute neutrophil count (ANC), hemoglobin (HGB) counts, platelet (PLT) counts, 2HG level, and DNA methylation level in UPN1825001 (**A**), UPN2463247 (**B**), UPN2297625 (**C**), UPN2620771 (**D**), and UPN2370759 (**E**). Line plots show mutation VAFs, ANC/HGB/PLT counts, and 2HG level. Blue shades represent BM blast counts. Violin plots show the methylation distribution. Violin plot shows the minimum, first quartile (Q1), median, third quartile (Q3), and maximum. Source data are provided as Source data files.

NCT01915498 (enasidenib for *IDH2*-mutated patients) and NCT02074839 (ivo-sidenib for *IDH1*-mutated patients). Selection of the studied patients was based on the sample availability alone. Bone marrow mononuclear cells were collected longitudinally (pretreatment, posttreatment, and relapse) from the trial participants and were subject for the analyses. Clinical response to the therapy was determined by the clinical investigators and followed the modified 2003 International Working Group criteria[26]. We defined responders as having overall response to the therapy, which included CR, CR with incomplete hematologic or platelet recovery, partial response, and MLFS. In MDS patients, hematologic improvement was also considered as response. Nonresponders were defined as having stable disease or progressive disease. Written informed consent for sample collection and analysis was obtained from all patients. The study protocols adhered to the Declaration of Helsinki and were approved by the Institutional Review Board at The University of Texas MD Anderson Cancer Center. Detailed information about the sample availability is shown in Fig. S14.

**Targeted deep sequencing and mutation caling**. We used a SureSelect custom panel of 295 genes or All Exon Panel v6. (Agilent Technologies, Santa Clara, CA; Table S2)[27]. For samples analyzed by exon panel, the analysis was restricted to the 295 genes. When available, clinical sequencing data were also used. Briefly, genomic DNA was extracted from bone marrow mononuclear cells using an Autopure extractor (QIAGEN/Gentra, Valencia, CA). DNAs were fragmented and bait-captured in solution according to manufacturer's protocols. Captured DNA

libraries were then sequenced using a HiSeq 2000 sequencer (Illumina, San Diego, CA) with 76 base pair paired-end reads. The median of median depth of the targeted regions was 323× (IQR: 201×–429×).

Raw-sequencing data from the Illumina platform were converted to a fastq format and aligned to the reference genome (hg19) using the Burroughs–Wheeler Aligner using MEM mode with following parameters: -k 31 -T 100 -t 8 -M. The aligned BAM files were subjected to mark duplication, realignment, and recalibration, using Picard and GATK with default parameters. Preprocessed BAM files were then analyzed to detect SNV and small insertions and deletions (indels), using MuTect and Pindel, respectively, against pooled unmatched normal sequences developed in house.

We performed a series of filtering and annotation to identify high-confidence driver mutations. First, variants with low-quality sequencing defined by (1) tumor coverage < 15×, (2) tumor allele frequency < 5%, and (3) normal allele frequency ≥ 1% and 0% for SNVs and INDELs, respectively, were filtered out. Second, only variants with obvious protein-coding change were kept for further analysis. Third, common polymorphisms with a population frequency of 0.14% in public variant databases, including the 1000 Genome Database, ESP6500 Database, dbSNP ver.129, and Exome Aggregation Consortium database, were removed. Finally, an hierarchical classification system was developed to assign confidence level for each remaining variant in order to facilitate the identification of putative oncogenic driver mutations. Specifically, each variant was classified based on the following hierarchical order and was assigned a confidence level corresponding to its rank in the system: (1) confirmed

**Table 3 A summary of available evidence for the association between molecular alterations and IDHi resistance, including the result from the current study.**

| Implicated mechanisms | References |
|---|---|
| Primary resistance | |
| Leukemia stemness | Current study |
| *RUNX1* mutations | Current study |
| *NRAS* mutations | Amatangelo et al.[9], Choe et al.[23] |
| *PTPN11* mutations | Choe et al.[23] |
| *FLT3* mutations (likely) | Choe et al.[23], current study |
| Acquired resistance | |
| Second-site *IDH* mutations | Intlekofer et al.[16], Choe et al.[23] |
| *IDH* homolog switch | Harding et al.[17], Quek et al.[15], Choe et al.[23], current study |
| RTK gene mutations (*RAS, FLT3, PTPN11, KIT*, and others) | Quek et al.[15], Choe et al.[23], current study |
| Differentiation gene mutations (*RUNX1, GATA2*, and *CEBPA*) | Quek et al.[15], Choe et al.[23], current study |
| *TET2* mutations | Choe et al.[23], current study |
| *BCOR* mutations | Quek et al.[15], current study |

somatic mutation based on COSMIC database version 81, (2) loss-of-function mutation such as splicing, stop-gain, stop-loss, and frameshift mutation in tumor suppressor genes, (3) variant which resides in the same position as a confirmed somatic mutation according to the COSMIC database, (4) recurrent variant which resides within three amino acids away from a confirmed somatic mutation according to the COSMIC database and was predicted to be damaging by in silico function prediction algorithms. The final annotated variant list was then further analyzed by manual inspection, in order to finalize the list of putative driver mutations.

**Methylation array profiling and data analysis**. DNA methylation analysis was performed using Illumina's Infinium MethylationEPIC assay (EPIC), according to the manufacturer's protocol[28]. Data analysis was conducted using the ChAMP algorithm[29] using default parameters. Briefly, The IDAT files were taken as input files and raw beta values were generated. Following initial quality check and probe filtering, including removing all SNP-related probes[30] and all probes located in chromosome X and Y, the data were normalized using the BMIQ method[31]. Differential methylation analysis was performed by using the limma algorithm[32].

**RNA sequencing and data analysis**. Strand specific RNA-sequencing libraries were constructed using the Illumina TruSeq RNA Access Library Prep Kit (Illumina, San Diego, CA), according to the manufacturer's protocol. Briefly, the double-stranded cDNA was hybridized to biotinylated, coding RNA capture probes. The resulting transcriptome-enriched library was sequenced by an Illumina HiSeq4000 using the 76 base pair paired-end configuration. Raw-sequencing data from the Illumina platform were converted to fastq files and aligned to the reference genome (hg19), using the STAR algorithm in single-pass mode with default parameters[33]. HTSeq-count was then utilized to generate the raw counts for each gene[34]. Raw counts were then analyzed by DESeq2 for data processing, normalization, and differential expression analysis, according to standard procedures[35].

**Single-cell targeted DNA sequencing and data analysis**. We performed a single-cell targeted DNA sequencing using Tapestri® platform (Mission Bio, South San Francisco, CA)[36,37]. Briefly, frozen bone marrow cells were thawed and resuspended with lysis buffer. Each cell was encapsulated into the microfluidic droplet, then was barcoded to label each cell differently. Barcoded samples were amplified using 50 primer pairs specific to the 19 mutated AML genes covering known disease-related hotspot loci (Supplementary Data 2). The pooled library was sequenced on an Illumina Miseq with 150-base pair paired end multiplexed runs. Fastq files generated from the MiSeq machine were processed using the Tapestri Analysis Pipeline (https://support.missionbio.com/hc/en-us/categories/360002512933-Tapestri-DNA-Pipeline) for adapter trimming, sequence alignment, barcode demultiplexing, and genotype and variant calling. Loom files generated by the pipeline were then analyzed by the in-house pipeline for variant annotation, filtering, and results visualization.

**NetBID activity analysis**. We performed NetBID2 analysis to identify hidden drivers of methylation-based cluster 1 and cluster 2 using RNA-seq data of baseline samples. We used normalized Log2 read count from RNA-seq of 26 cluster 1 and 13 cluster 2 samples as input to generate networks using SJARACNe[38]. We used P value < 0.005, log FC > 0.04, size ≤ 1000 and size ≥ 30 to select drivers.

**LSC17 score calculation**. LSC17 score was calculated using RNA-seq data from baseline samples according to the equation published previously[22]: LSC17 score = $(DNMT3B \times 0.0874) + (ZBTB46 \times -0.0347) + (NYNRIN \times 0.00865) + (ARHGAP22 \times -0.0138) + (LAPTM4B \times 0.00582) + (MMRN1 \times 0.0258) + (DPYSL3 \times 0.0284) + (KIAA0125 \times 0.0196) + (CDK6 \times -0.0704) + (CPXM1 \times -0.0258) + (SOCS2 \times 0.0271) + (SMIM24 \times -0.0226) + (EMP1 \times 0.0146) + (NGFRAP1 \times 0.0465) + (CD34 \times 0.0338) + (AKR1C3 \times -0.0402) + (GPR56 \times 0.0501)$.

**Statistical analysis**. The Chi-square or Fisher's exact test was used to assess statistical differences in categorical variables and odds ratio to evaluate the strength of association. The Mann–Whitney $U$ test or Student's $t$ test was used to analyze differences in continuous variables. Multiple hypothesis testing was corrected by Benjamini–Hochberg method. ROC curve as well as AUROC value were generated by pROC R package. A multivariate logistic regression model was performed to examine the relationship between CR variable and the predictors of LSC17 score, RUNX1 mutation, RAS/RTK mutation, and ELN high-risk cytogenetics. All applicable tests were two-sided and $P$ value < 0.05 was considered as statistical significance. Statistical analyses were performed within the analytic software described above or by R computing software (ver. 3.3.2).

**Reporting summary**. Further information on research design is available in the Nature Research Reporting Summary linked to this article.

## Data availability

Raw targeted deep sequencing and single-cell targeted DNA-seq data generated in this study have been deposited in the SRA with the accession number PRJNA713342. Raw methylation and RNA-Seq data generated in this study have been deposited in the GEO with the accession number GSE153349. The full list of detected baseline driver mutations was shown in Supplementary Data 3. Source data are provided with this paper.

## Code availability

The custom codes that support the findings of this study is available in GitHub (https://doi.org/10.5281/zenodo.4606803).

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

## Acknowledgements

This study was supported in part by the Cancer Prevention Research Institute of Texas (grant R120501 to P.A.F.), the Welch Foundation (grant G-0040 to P.A.F.), the UT System STARS Award (grant PS100149 to P.A.F.), Khalifa Scholar Award (to K.T.), Physician Scientist Program at MD Anderson (to K.T.), V Foundation Lloyd Family Clinical Scholar Award (to C.D.D.), Charif Souki Cancer Research Fund (to H.K.), MD Anderson Cancer Center Leukemia SPORE P50 CA100632 (to H.K.), MD Anderson Cancer Center Support Grant (NIH P30 CA016672), Lynda Hill Foundation (to P.A.F.), JSPS Overseas Research Fellowship (to T.T.), and by generous philanthropic contributions to MD Anderson's Moon Shot Program for AML/MDS Platform (to H.K., G.G.M., and K.T.). We thank Sunita Patterson at Department Scientific Publications at MD Anderson for providing scientific editing of the manuscript.

## Author contributions

Conception and design of the study was performed by K.T., C.D.D., and P.A.F. K.T. and P.A.F. supervised the study and provided financial support. C.D.D. and H.K. led all the clinical trials relevant to this study. Sample collection and data assembly were performed by T.T., K.P., K.J.M., B.W., G.L., M.F., J.M., L.L., C.G., E.T., T.K., G.G.-M., E.J., F.R., K.B., M.K., H.K., and K.T. F.W., K.M., K.F., Y.Y., S.X., J.Z., and K.T. performed data analysis and interpretation. F.W. and K.T. wrote the manuscript and revised based on the input from all other authors. All authors approved the final version of the manuscript.

## Competing interests

K.T. receives advisory and consultancy fee from Celgene, Novartis, GSK, Symbio Pharmaceuticals, and Kyowa Hakko Kirin. K.M.B. and M.F. were Celgene employee. B.W. and G.L. were Agios employee. C.D.D. receives research support (to institution) from Abbvie, Agios, Calithera, Cleave, BMS/Celgene, Daiichi-Sankyo, Forma, Loxo, and ImmuneOnc, and is among the Consultant/Advisory Boards at Abbvie, Agios, Celgene/BMS, ImmuneOnc, Novartis, Takeda, and Aprea. C.D.D. is a scientific advisor with stock options from Notable Labs. K.N.B. is a consultant for Iterion Therapeutics. E.J. receives research support (to institution) and consultancy fee from AbbVie, Amgen, Adaptive biotechnologies, Ascentage, BMS/Celgene, Genentech, Pfizer, and Takeda. H.K. receives research grants from AbbVie, Amgen, Ascentage, BMS, Daiichi-Sankyo, Immunogen, Jazz, Novartis, Pfizer and Sanofi, and honoraria from AbbVie, Actinium (Advisory Board), Adaptive Biotechnologies, Amgen, Apptitude Health, BioAscend, Daiichi-Sankyo, Delta Fly, Janssen Global, Novartis, Oxford Biometical, Pfizer and Takeda. F.R. is member of advisory boards for Celgene, BMS, and Agios, and receives honoraria from them. K.P. receives consultancy fee from Novartis. T.K. receives consulting fee from AbbVie, Agios, Daiichi-Sankyo, Genentech, Jazz Pharmaceuticals, Liberum, Novartis, Pfizer, Sanofi-Aventis, and receives research support (to institution) from AbbVie, Amgen, BMS, Genentech, Jazz Pharmaceuticals, Pfizer, Cellenkos, Ascentage, Genfleet, Astellas, and AstraZeneca, and honoraria from Genzyme. All other authors declare no competing interests.
