## [Peer Review File · Nature Communications]

REVIEWER COMMENTS

Reviewer #1 (Remarks to the Author):

In this paper, “Evolution of AML genome and epigenome with IDH inhibitors and their association with clinical response and resistance,” Wang, Morita, DiNardo, and colleagues performed various genomic analyses (DNA sequencing, RNA sequencing and cytosine methylation profiling) in longitudinally collected specimens from 68 IDH1/IDH2-mutant AML patients treated with IDH inhibitors (IDHi), and then examined the evolution of AML genome and epigenome during the therapy and its association with clinical response and relapse. A wide range of clinical responses were observed, including reduced response to the drugs when a co-occurrence of mutations in RUNX1/CEBPA or RAS-RTK pathway genes was observed.

Also, acquired mutations in BCOR, reciprocal IDH gene, and TET2 were enriched at relapse. The authors used exome sequencing at plenty of depth (393x), standard methylation arrays (EPIC), stranded library prep for RNA-sequencing, and single-cell Tapestry variant profiling, to generate broad profiles. While the results are interesting and represent a nice cohort that is needed in the field, the significance of these enrichments is constrained by the sample sizes, and the authors are careful about this in their descriptions. Overall the work is relevant and useful to the field, but could benefit from a few more clarifications and analyses:

- 1) The authors note that sample selection was based on sample availability alone, and so they need to ensure there are not any biases from these selections for their statistical calculations for enrichments.
- 2) The authors found additional evidence of IDH isoform switching, which is intriguing, but the same analysis should be repeated across all captured genes to help establish the null distribution of such events (likely zero, but this should be confirmed) and also to rule out any technical artifacts from the amplification processes.
- 3) Given the impact of clonal hematopoiesis as a function of age, did the authors observe any difference in known CH genes or co-variables as a function of age? Or was any of this evident in the single-cell variant data?
- 4) Data is stated to be available upon request, but the data needs to be deposited into a public archive (e.g. dbGAP).
- 5) Code is stated to be available upon request, but this needs to be available with the paper or it is impossible to know how or if the filters and methods used for the analysis are appropriate or ideal.
- 6) In a related note, the computational methods need a bit more clarity. For example, they mention which aligners they used, but without the key details (e.g. STAR, two-pass one single-pass, it is not specified) or command-line parameters.
- 7) The authors note the CIMP results, as expected, but the heterogeneity of methylation in AML is a possible additional clinical phenotype that should be explored, and prior relevant work that has already shown this should be cited, such as <https://www.nature.com/articles/nm.4125> and <https://pubmed.ncbi.nlm.nih.gov/25260792/>. For example, genes where the methylation changes at relapse may relate to differentially expressed genes, and this disruption can lead to increased expression variance, which should be examined here.
- 8) Do the authors see evidence of sub-clonality at low VAF for the single-cell data? This also might be present at low levels, but would be missed by the default variant calling parameters for most pipelines.
- 9) Did the variants observed in isoform switching also show up in the RNA-seq data? There is a GATK RNA-seq variant pipeline that could answer this question readily and should be run.

Thank you.

Reviewer #2 (Remarks to the Author):

The manuscript entitled Evolution of AML genome and epigenome with IDH inhibitors and their association with clinical response and resistance by Wang and co-workers describes the detailed analyses of 68 IDH mutant AML patients receiving various IDH inhibitors. The authors analyzed the AML samples the genome, transcriptome and epigenome of bulk populations and at single cell level. A subset of patients was longitudinally analyzed during the course of their disease.

We all know that AML is a heterogeneous disease, even in selected subsets of AML, such as AML with mutations in either IDH1 or IDH2. This makes any study in these mixed groups challenging. In this study a relatively large cohort of hematologic malignancies was studied. However, the cohort did not only include AML, but also cases with MDS and CMML (n=6). This may even add to variability in results. Furthermore, although the majority of patients were treated with the IDH inhibitors ivosidenib (n=22) and enasidenib (n=38), other inhibitors were administered in the remaining cases. One could wonder if the use of these other inhibitors may result in different outcomes or dynamics. Although the analyses of clonal evolution of IDH mutant AML (during treatment with inhibitors) is an important task for future research, it may require very large cohorts to recognize relevant patterns. In my opinion the manuscript by Wang and co-workers is interesting but is rather descriptive.

It is not entirely clear if all diagnosis, follow-up and relapse samples were sequenced with the Sure select custom panel. With a median depth of app. 400 small subclones may be missed.

It is not easy to follow which samples were used for which analyses and why. I am also referring to the differences in malignancies and inhibitors. Also, which relapse cases were selected for the single cell analyses and why? Likewise, when discussing the different patterns in clonal dynamics it is unclear which and how many cases followed each pattern.

The authors indicate that there was no association with clearance of mutations and response. Were the mutations in the cases in which the VAF of the AML mutations remained unchanged related to clonal hematopoiesis? This could suggest that these may not have been relevant with regard to IDH inhibitor response. For instance, the IDH2 mutation in figure 2E seems more like clonal hematopoiesis and not related to the diagnosis/relapse AML.

Several findings were confirmatory, as indicated by the authors as well, for instance the association of co-occurring RAS mutations and response.

The expression analyses is again quite descriptive and does not seem to add much.

We would like to first thank the reviewers for their critical assessment of our manuscript and their constructive feedback. We are most grateful for the comments which have tremendously helped us to improve considerably the manuscript. Our point by point response is outlined below. We thank you in advance for your attention and hope you will find our manuscript suitable for publication in *Nature Communications*.

Reviewer #1: In this paper, “Evolution of AML genome and epigenome with IDH inhibitors and their association with clinical response and resistance,” Wang, Morita, DiNardo, and colleagues performed various genomic analyses (DNA sequencing, RNA sequencing and cytosine methylation profiling) in longitudinally collected specimens from 68 IDH1/IDH2-mutant AML patients treated with IDH inhibitors (IDHi), and then examined the evolution of AML genome and epigenome during the therapy and its association with clinical response and relapse. A wide range of clinical responses were observed, including reduced response to the drugs when a co-occurrence of mutations in RUNX1/CEBPA or RAS-RTK pathway genes was observed. Also, acquired mutations in BCOR, reciprocal IDH gene, and TET2 were enriched at relapse. The authors used exome sequencing at plenty of depth (393x), standard methylation arrays (EPIC), stranded library prep for RNA-sequencing, and single-cell Tapestry variant profiling, to generate broad profiles. While the results are interesting and represent a nice cohort that is needed in the field, the significance of these enrichments is constrained by the sample sizes, and the authors are careful about this in their descriptions. Overall the work is relevant and useful to the field, but could benefit from a few more clarifications and analyses:

- 1) The authors note that sample selection was based on sample availability alone, and so they need to ensure there are not any biases from these selections for their statistical calculations for enrichments.*

RE: Thanks for the comment. We have collected the clinical information of the patients who were not analyzed in this study (due to the non-availability of samples) and compared the clinical characteristics with analyzed patients (**Table below**). Based on this analysis, the patients analyzed in this study were significantly older, had lower neutrophil counts and higher bone marrow blast counts than the patients not analyzed in the study. Also, the analyzed cohort had included more responders than excluded cohorts. These differences suggest that our cohort might have been underpowered to detect molecular predictors for primary resistance to IDH inhibitor, which is consistent with our data that *RUNX1* mutations were the only statistically significant marker associated with primary resistance, whereas *RAS* mutations, *FLT3* mutations, and *TP53* mutations did not reach statistical significance, and this might be associated with the under-enrichment of non-responders in our cohort. We acknowledged this limitation and added the statement in the discussion (**page 15**). Also, we have added the data comparing clinical characteristics between analyzed patients and not-analyzed patients (**Supp Table S1, Page 5**)

Table.

	Analyzed (N=68)		Not-analyzed (N=80)		p
	median	IQR	median	IQR	
WBC	1.5	0.9-2.3	1.8	1.0-8.0	0.084
ANC	0.2	0.1-0.6	0.5	0.2-1.7	0.006
HGB	9.4	9.8-10.1	9.4	8.8-10.2	0.965
PLT	33	20-64	35	19-71	0.913
BM blast, %	39	18-63	27	9-56	0.045
Age	72	65-77	67	58-73	0.01
	No.	%	No.	%	
Karyotype					1
intermediate	46	70	55	70	
poor	20	30	24	30	
Sex					0.498
female	28	41	28	35	
male	40	59	52	65	
Drug					<0.001
AG120	22	32	33	41	
AG221	38	56	18	23	
AG881	1	2	17	21	
IDH305	7	10	12	15	
Best Response					
CR	17	25	5	6	
CRi	1	2	1	1	
CRp	7	10	4	5	
MLFS	9	13	5	6	
PR	3	4	2	3	
HI	1	2	2	3	
SD	28	41	29	36	
PD	1	2	18	23	
Not done	1	2	14	18	
Responder	38	57	19	29	0.002
Non-Responder	29	43	47	71	

ND – Student's t-test; non-ND – Mann–Whitney U test

WBC – non-ND; ANC– non-ND; HGB – ND; PLT– non-ND; BM blast, %– non-ND; Age– non-ND

Abbreviations: ND, normal distribution

2) *The authors found additional evidence of IDH isoform switching, which is intriguing, but the same analysis should be repeated across all captured genes to help establish the null distribution of such events (likely zero, but this should be confirmed) and also to rule out any technical artifacts from the amplification processes.*

RE: Thank you for this remark. We believe that there is a misunderstanding about the term “isoform switching”. The term “isoform switching” was used to describe a phenomenon, in which emergence of a mutation in a homologous *IDH* gene was observed during the inhibition of another *IDH* gene mutation. For example, if the patient with *IDH2*-mutated AML was treated with *IDH2* inhibitor, and *IDH1* mutation was arising at the relapse, then this phenomenon was described as “isoform switch”. We have to agree with the reviewer that this description was somewhat confusing, and the reviewer might have interpreted as an emergence of a different isoform of the same gene (i.e., distinctly spliced transcripts). The reason why we used the term “isoform switch” for this description was to maintain the consistency with the prior publication that described the same phenomenon (Harding et al. Cancer Discovery “Isoform switching as a mechanism of acquired resistance to mutant isocitrate dehydrogenase inhibition”). We hope this clarifies the reviewer’s concern about the isoform switching phenomenon. To avoid additional confusion from the general readership, we added clarification in the paper (**page 4**)

3) *Given the impact of clonal hematopoiesis as a function of age, did the authors observe any difference in known CH genes or co-variables as a function of age? Or was any of this evident in the single-cell variant data?*

RE: We examined three common CH genes *DMNT3A/TET2/ASXL1* (DTA) in baseline samples and compared the age of cases with and without the mutations using student t test. There were no significant age differences between DTA vs non-DTA case (**Letter Figure 1**). The studied cohort is already enriched for elderly population (median age of the entire cohort is 72), which likely is the reason why we did not see the difference between DTA+ and DTA- cases.

Letter Figure 1.

	With DTA (N=50)	Without -DTA (N=17)
Median Age	69.08 (IQR: 63.75-77)	71.24 (IQR: 63-80)

- 4) *Data is stated to be available upon request, but the data needs to be deposited into a public archive (e.g. dbGAP).*

RE: We apologize for the delay in the data deposition. We have completed the submission of the methylation and RNA-Seq data to GEO with the accession number **GSE153349**. The full list of the detected baseline mutations was provided in **Table S4**.

- 5) *Code is stated to be available upon request, but this needs to be available with the paper or it is impossible to know how or if the filters and methods used for the analysis are appropriate or ideal.*

RE: We apologize for the inconvenience. We have added the following description for targeted deep sequencing data analysis section in supplementary methods. Codes for the annotation of variants generated by targeted deep sequencing as well as variant extraction and filtering of single cell sequencing data were provided in GitHub (https://github.com/farmerkingwf/IDH_codes.git).

Raw sequencing data from the Illumina platform were converted to a fastq format and aligned to the reference genome (hg19) using the Burroughs-Wheeler Aligner (BWA). BWA is using MEM mode with following parameters: -k 31 -T 100 -t 8 -M. The aligned BAM files were subjected to mark duplication, re-alignment, and re-calibration using Picard and GATK with default parameters (<https://www.broadinstitute.org/gatk/guide/best-practices?bpm=DNaseq>, last accessed 9/29/2016). Preprocessed BAM files were then analyzed to detect single nucleotide variants (SNV) and small insertions and deletions (indels) using MuTect (<https://pubmed.ncbi.nlm.nih.gov/19451168/>) and Pindel (<https://pubmed.ncbi.nlm.nih.gov/19561018/>) algorithms, respectively. We performed a series of filtering and annotation to identify high-confidence driver mutations. First, variants with low quality sequencing data were filtered out.

Specifically, variants matching one or more of the following criteria were considered of low quality and therefore filtered out from further analysis: 1) tumor coverage < 15X, 2) tumor allele frequency < 5%, and 3) normal allele frequency >= 1% and 0% for SNVs and INDELS, respectively. Second, only variants which would introduce an obvious protein-coding change were kept for further analysis. Specifically, variant with an ANNOVAR annotation of non-synonymous, stop-gain, stop-loss, splicing, frameshift insertion, frameshift deletion, nonframeshift insertion or nonframeshift deletion was considered to be able to introduce an obvious protein-coding change and were therefore kept for further analysis. Third, common polymorphisms were removed to reduce the load of possible germline contamination due to the absence of matched normal. Specifically, a series of public variant database including the 1000 Genome Database (<http://www.1000genomes.org/>), ESP6500 Database (<http://evs.gs.washington.edu/EVS/>), dbSNP ver.129 (<http://www.ncbi.nlm.nih.gov/SNP/>), and Exome Aggregation Consortium database (<http://exac.broadinstitute.org/>), were utilized. Variant with a population frequency of 0.14% or more in any of the databases was considered possible germline polymorphism and was therefore removed from further analysis. Finally, a hierarchical classification system was developed to assign confidence level for each remaining variant in order to facilitate the identification of putative driver mutations. Specifically, each variant was classified based on the following hierarchical order and was assigned a confidence level corresponding to its rank in the system: 1) Confirmed somatic mutation based on COSMIC database (version 81), 2) loss-of-function mutation such as splicing, stop-gain, stop-loss and frameshift mutation in tumor suppressor genes, 3) variant which resides in the same position as a confirmed somatic mutation according to the COSMIC database, 4) recurrent variant which resides within three amino acids away from a confirmed somatic mutation according to the COSMIC database and was predicted to be damaging by in-silico function prediction algorithms. The final annotated variant list was then further analyzed by manual inspection in order to identify putative driver mutation.

- 6) *In a related note, the computational methods need a bit more clarity. For example, they mention which aligners they used, but without the key details (e.g. STAR, two-pass one single-pass, it is not specified) or command-line parameters.*

RE: We apologize that our description was not clear enough. STAR is running in single-pass mode with default parameters. BWA is using MEM mode with following parameters: -k 31 -T 100 -t 8 -M. All additional software including GATK, Mutect, Pindel, HTSeq-Count, ChAMP, and DESeq2 was used with default parameters or following standard procedures. The information has been updated in the revised manuscript (**page 18**).

- 7) *The authors note the CIMP results, as expected, but the heterogeneity of methylation in AML is a possible additional clinical phenotype that should be explored, and prior relevant work that has already shown this should be cited, such as <https://www.nature.com/articles/nm.4125> and <https://pubmed.ncbi.nlm.nih.gov/25260792/>. For example, genes where the methylation changes at relapse may relate*

to differentially expressed genes, and this disruption can lead to increased expression variance, which should be examined here.

RE: We greatly appreciate this comment by the reviewer. This suggestion prompted us to conduct detailed analyses of methylation heterogeneity and its association with clinical phenotype among our samples, which led to significant updates in the manuscript detailed in **Page 6-7 and Figure 2-3**. Briefly, k-means consensus clustering of DNA methylation profiles on pre-treatment samples identified two distinct methylation cluster (Cluster 1 as relatively hypomethylated cluster and Cluster 2 as relatively hypermethylated cluster). Notably, Cluster 2 (hypermethylated cluster) was associated with significantly inferior response to IDHi. Also, Cluster 2 was associated with significantly less *DNMT3A* mutation, which is consistent with the relatively hypermethylated state of the cluster (co-occurrence of *DNMT3A* and *IDH* mutations lead to methylation antagonism) (Glass et al., 2017). Gene expression analysis between the two clusters showed that leukemia stemness genes are significantly upregulated in Cluster 2. Finally, we also show that stemness score (LSC17) (Ng et al., 2016) predicted response to IDHi better than other molecular markers such as *RUNX1* mutations, *RAS* mutations or cytogenetic risks. We believe that these data add novel insights into molecular mechanism of resistance to IDHi and greatly appreciate the reviewer for making this suggestion. As far as the two papers the reviewer suggested, we could not use the same pipeline (Methclone) in our analysis because our methylation data was based on illumina EPIC microarray and not bisulfite sequencing. However, we thank the reviewers for bringing up these important papers to our attention.

8) *Do the authors see evidence of sub-clonality at low VAF for the single-cell data? This also might be present at low levels, but would be missed by the default variant calling parameters for most pipelines.*

RE: The reviewer is absolutely correct that single-cell DNA sequencing will provide evidence for clonal substructure in AML. This is shown in Figure 5E-F where single-cell data shows subclonal architecture in 2 samples. We also would like to highlight our separate research that was recently published in Nature Communications, where we described subclonal architectures in a large cohort of AML samples (Morita et al., 2020). Bulk sequencing can not prove mutation co-occurrence among low VAF mutations and this is one of the unique aspects of single-cell DNA sequencing.

9) *Did the variants observed in isoform switching also show up in the RNA-seq data? There is a GATK RNA-seq variant pipeline that could answer this question readily and should be run.*

RE: Related to the comment #2, we think that there is a misunderstanding about the concept of “isoform switching”. Here, the isoform switching refers to the emergence of the mutations in a homologous IDH genes driving relapse but not the emergence of differently spliced isoform in the same IDH gene. We believe that the reviewer is

asking to check differently spliced isoforms of IDH genes in the RNA-seq data. But this is not what the isoform switching refers to in this manuscript.

Reviewer #2: The manuscript entitled Evolution of AML genome and epigenome with IDH inhibitors and their association with clinical response and resistance by Wang and co-workers describes the detailed analyses of 68 IDH mutant AML patients receiving various IDH inhibitors. The authors analyzed the AML samples the genome, transcriptome and epigenome of bulk populations and at single cell level. A subset of patients was longitudinally analyzed during the course of their disease. We all know that AML is a heterogenous disease, even in selected subsets of AML, such as AML with mutations in either IDH1 or IDH2. This makes any study in these mixed groups challenging. In this study a relatively large cohort of hematologic malignancies was studied.

10) However, the cohort did not only included AML, but also case with MDS and CMML (n=6). This may even add to variability in results. Furthermore, although the majority of patients were treated with the IDH inhibitors ivosidenib (n=22) and enasidenib (n=38), other inhibitors were administered in the remaining cases. One could wonder if the use of these other inhibitors may result in different outcomes or dynamics.

RE: We agree with the reviewer that the heterogeneities in diagnosis and drug treatment might have confounded the results. As far as the drug treatment is concerned, our cohort included 7 patients (10% of the entire cohort) treated with IDH305 (IDH1 inhibitor) and one patient with AG881 (pan-IDH inhibitor). We compared clinical characteristics of patients treated with AG221, AG120, or IDH305, and did not find significant differences among the treatment groups (**Table below**, we did not compare AG881 since it was only 1 patient). While clinical trials of IDH305 is still ongoing, results from Phase 1 trial presented by DiNardo et al. demonstrated that 33% overall response rate in AML patients, which is comparable to ivosidenib (DiNardo et al., 2016). Also, clonal dynamics of *IDH1* mutation and co-occurring mutations in IDH305-treated samples did not show particularly unique dynamics (figure below). Collectively, we believe that inclusion of IDH305 treated samples does not significantly affect the overall conclusion of our paper. In regard to the heterogeneity in diagnosis, our cohort included 5 MDS patients and 1 CMML patient, which together consisted of 9% of the cohort. Previous studies have shown that *IDH*-mutated MDS have propensity for AML progression. Makishima and colleagues have previously defined *IDH1* and *IDH2* mutations as type 1 mutations in MDS, which were strongly associated with AML transformation (Makishima et al., 2017). Also, gene expression profiling of MDS samples showed that *IDH1/IDH2*-mutated MDS were enriched with immature myeloid progenitor (IMP) signature which was also associated with AML transformation (Shiozawa et al., 2017). These data suggest that *IDH*-mutated MDS has some biological similarity to AML.

Clinical trials of IDH inhibitors in MDS/CMML patients are currently ongoing. Results from the trials have not been published in peer-reviewed medical journals but based

on the abstract presented at American Society of Hematology in 2019 (Richard-Carpentier et al., 2019), 25 MDS patients have been treated by a multicenter phase II trial of enasidenib and demonstrated 67% overall response to the drug. The response rate in MDS is comparable to the overall response rate in AML treated with enasidenib or ivosidenib (DiNardo et al., 2018; Stein et al., 2017). Taken together, we believe that small numbers of MDS/CMML patients in this study will not significantly affect the findings in our paper. That being said, these heterogeneities are important limitations of our study. We added a statement acknowledging this in the paper (**page 15-16**)

Clinical characteristics comparing patients treated with AG221, AG120, and IDH305

	AG120 (N=22)		AG221 (N=38)		IDH305 (N=7)		p
	median	IQR	median	IQR	median	IQR	
WBC	1.3	1.0-2.4	1.6	0.8-2.4	1.3	0.9-1.7	0.773
ANC	0.1	0.0-0.4	0.4	0.1-0.6	0.2	0.1-0.3	0.209
HGB	9.6	9.1-10.5	9.2	8.5-9.9	10.2	9.7-11.6	0.003
PLT	33	23-54	36	16-61	30	21-82	0.984
BM blast, %	39	21-55	43	17-66	36	18-51	0.848
Age	70	60-76	73	61-77	74	72-76	0.382
	No.	%	No.	%	No.	%	
Karyotype							0.674
intermediate	13	59	27	71	5	71	
poor	8	36	10	26	2	29	
Sex							0.817
female	8	36	17	45	3	43	
male	14	64	21	55	4	57	
Diagnosis							0.719
AML	20	91	35	92	7	100	
MDS/CMML	2	9	3	8	0	0	

Clonal dynamics of IDH305 treated patients

11) It is not entirely clear if all diagnosis, follow-up and relapse samples were sequenced with the Sure select custom panel.

RE: We apologize that the description was not so clear. All longitudinal samples were sequenced by the same 300 gene panel. We clarified this in the revised manuscript (**page 17**)

12) With a median depth of app. 400 small subclones may be missed.

RE: The reviewer is absolutely correct that the median 400x NGS sequencing will miss very small subclones. Overall, our conservative estimate of sequencing sensitivity is at VAF of 5%. We used relatively conservative cut off because conservative cut-off has been shown to be better for clinical correlation analysis

(Papaemmanuil et al., 2013). We acknowledged this limitation in the discussion (**Page 16**)

13) It is not easy to follow which samples were used for which analyses and why. I am also referring to the differences in malignancies and inhibitors.

RE: Our intention was to analyze all the samples (including longitudinal) with all the platforms (DNA seq, RNA seq, methylation array). However, due to the limited amount of sample availability, not all samples were analyzed by all the platform. The sample-analysis matchup table is shown in **Figure S14**. We acknowledged this as one of the limitations of our study (**page 15**)

14) Also, which relapse cases were selected for the single cell analyses and why?

RE: Ideally, we wanted to analyze all the relapse cases with single-cell analysis but the analysis requires viable cells and because of limited sample availability and also due to the limitation in genes covered by the single-cell DNA sequencing, we were only able to analyze two cases, Case UPN2394529 and case UPN2297707. However, one case had relapse-associated mutations independent of IDH clone, and the other case had relapse-associated mutations as part of IDH clone. Therefore, although the number of analyzed cases is low, we believe it highlights the distinct patterns of clonal relationship of relapse-driving mutations and IDH mutations.

15) Likewise, when discussing the different patterns in clonal dynamics it is unclear which and how many cases followed each pattern.

RE: Thank you for this important question. We have explicitly stated the number of samples we were able to analyze for this co-evolution patterns (**page 11 -12**). Since this analysis required a complete set of multi-dimensional data, we were only able to analyze this in 16 patients of which 5 were responders and 11 were non-responders. Therefore, in Figure 6, we have shown all the patterns identified in 5 responders. For 11 non-responders, we have added the description in **page 12** and **Figure S13**.

16) The authors indicate that there was no association with clearance of mutations and response. Were the mutations in the cases in which the VAF of the AML mutations remained unchanged related to clonal hematopoiesis? This could suggest that these may not have been relevant with regard to IDH inhibitor response. For instance, the IDH2 mutation in figure 2E seems more like clonal hematopoiesis and not related to the diagnosis/relapse AML.

RE: Thank you for this important question. As the reviewer rightly points out, clonal hematopoiesis mutations can persist in remission after chemotherapies, which has been implicated to have little impact on AML prognosis. In fact, our lab was one of the labs systematically analyzed this in a large cohort of AML patients (Jongen-Lavrencic et al., 2018; Morita et al., 2018).

However, we do not believe that persistent mutations in IDHi treated cases represent clonal hematopoiesis. Clinical response to IDH inhibitors is facilitated by terminal differentiation of IDH-mutant leukemia cells, and not through inducing apoptosis. Many previous studies have shown this unique MOA of IDHi and that differentiated cells continue to carry *IDH* mutations (reviewed in Introduction [page 3]) (Kernytsky et al., 2015; Quek et al., 2018; Wang et al., 2013; Yen et al., 2017). Therefore, it stands to reason that VAF of IDH mutations do not change in majority of the patients achieving complete remission. However, this is different from patients having persistent clonal hematopoiesis after achieving remission with cytotoxic chemotherapy.

One way to distinguish this is to analyze the dynamics of co-occurring mutations. In cases with persistent clonal hematopoiesis, co-occurring mutations that are specific to leukemic blasts (e.g., *NPM1*, *FLT3*, *N-K-RAS*, *CEBPA*, and others) are often cleared while clonal hematopoiesis mutations persist (*DNMT3A*, *TET2*, *ASXL1* and others) (Jongen-Lavrencic et al., 2018; Morita et al., 2018). In contrast, many of the co-occurring mutations remains stable after IDHi therapy even in patients who demonstrate clinical response (**Figure 6**). These patterns of clonal dynamics suggest that persistent mutations after IDHi therapy do not necessarily represent clonal hematopoiesis but rather represent mutant cells that are morphologically differentiated. This is somewhat similar to the phenomenon seen in acute promyelocytic leukemia (APL) in which retinoic acid therapy facilitates differentiation of promyelocytic cells but *PML-RARA* rearrangement continues to be detected in remission samples.

17) Several findings were confirmatory, as indicated by the authors as well, for instance the association of co-occurring RAS mutations and response.

RE: We agree with the reviewer's comment that some of our findings are consistent with previous reports and acknowledged this in the discussion (**page 12**). We believe that the new data in the revised manuscript (related to Reviewer #1, comment #7, **Figure 2-3**) adds novelty in our paper. Also, we have added a new figure summarizing the available evidence so far for the molecular mechanisms of resistance to IDHi (**Figure 6F**).

18) The expression analysis is again quite descriptive and does not seem to add much.

RE: Thank you for this remark. Based on the suggestion by reviewer #1 comment #7, we now identified gene expression signature with stemness is associated with resistance to IDHi (**page 6-7, Figure 2-3**). We believe that these new data provide novel insights into the resistance mechanism of IDHi in AML.

References

DiNardo, C.D., Schimmer, A.D., Yee, K.W.L., Hochhaus, A., Kraemer, A., Carvajal, R.D., Janku, F., Bedard, P., Carpio, C., Wick, A., *et al.* (2016). A Phase I Study of IDH305 in Patients with Advanced Malignancies Including Relapsed/Refractory AML and MDS That Harbor IDH1R132 Mutations. *Blood* 128, 1073-1073.

DiNardo, C.D., Stein, E.M., de Botton, S., Roboz, G.J., Altman, J.K., Mims, A.S., Swords, R., Collins, R.H., Mannis, G.N., Pollyea, D.A., *et al.* (2018). Durable Remissions with Ivosidenib in IDH1-Mutated Relapsed or Refractory AML. *N Engl J Med* 378, 2386-2398.

Glass, J.L., Hassane, D., Wouters, B.J., Kunimoto, H., Avellino, R., Garrett-Bakelman, F.E., Guryanova, O.A., Bowman, R., Redlich, S., Intlekofer, A.M., *et al.* (2017). Epigenetic Identity in AML Depends on Disruption of Nonpromoter Regulatory Elements and Is Affected by Antagonistic Effects of Mutations in Epigenetic Modifiers. *Cancer Discov* 7, 868-883.

Jongen-Lavrencic, M., Grob, T., Hanekamp, D., Kavelaars, F.G., Al Hinai, A., Zeilemaker, A., Erpelinck-Verschueren, C.A.J., Gradowska, P.L., Meijer, R., Cloos, J., *et al.* (2018). Molecular Minimal Residual Disease in Acute Myeloid Leukemia. *N Engl J Med* 378, 1189-1199.

Kernytsky, A., Wang, F., Hansen, E., Schalm, S., Straley, K., Gliser, C., Yang, H., Travins, J., Murray, S., Dorsch, M., *et al.* (2015). IDH2 mutation-induced histone and DNA hypermethylation is progressively reversed by small-molecule inhibition. *Blood* 125, 296-303.

Makishima, H., Yoshizato, T., Yoshida, K., Sekeres, M.A., Radivoyevitch, T., Suzuki, H., Przychodzen, B., Nagata, Y., Meggendorfer, M., Sanada, M., *et al.* (2017). Dynamics of clonal evolution in myelodysplastic syndromes. *Nat Genet* 49, 204-212.

Morita, K., Kantarjian, H.M., Wang, F., Yan, Y., Bueso-Ramos, C., Sasaki, K., Issa, G.C., Wang, S., Jorgensen, J., Song, X., *et al.* (2018). Clearance of Somatic Mutations at Remission and the Risk of Relapse in Acute Myeloid Leukemia. *J Clin Oncol* 36, 1788-1797.

Morita, K., Wang, F., Jahn, K., Hu, T., Tanaka, T., Sasaki, Y., Kuipers, J., Loghavi, S., Wang, S.A., Yan, Y., *et al.* (2020). Clonal evolution of acute myeloid leukemia revealed by high-throughput single-cell genomics. *Nature Communications* 11, 5327.

Ng, S.W., Mitchell, A., Kennedy, J.A., Chen, W.C., McLeod, J., Ibrahimova, N., Arruda, A., Popescu, A., Gupta, V., Schimmer, A.D., *et al.* (2016). A 17-gene stemness score for rapid determination of risk in acute leukaemia. *Nature* 540, 433-437.

Papaemmanuil, E., Gerstung, M., Malcovati, L., Tauro, S., Gundem, G., Van Loo, P., Yoon, C.J., Ellis, P., Wedge, D.C., Pellagatti, A., *et al.* (2013). Clinical and biological

implications of driver mutations in myelodysplastic syndromes. *Blood* 122, 3616-3627; quiz 3699.

Quek, L., David, M.D., Kennedy, A., Metzner, M., Amatangelo, M., Shih, A., Stoilova, B., Quivoron, C., Heiblig, M., Willekens, C., *et al.* (2018). Clonal heterogeneity of acute myeloid leukemia treated with the IDH2 inhibitor enasidenib. *Nat Med* 24, 1167-1177.

Richard-Carpentier, G., DeZern, A.E., Takahashi, K., Konopleva, M.Y., Loghavi, S., Masarova, L., Alvarado, Y., Ravandi, F., Montalban Bravo, G., Naqvi, K., *et al.* (2019). Preliminary Results from the Phase II Study of the IDH2-Inhibitor Enasidenib in Patients with High-Risk IDH2-Mutated Myelodysplastic Syndromes (MDS). *Blood* 134, 678-678.

Shiozawa, Y., Malcovati, L., Galli, A., Pellagatti, A., Karimi, M., Sato-Otsubo, A., Sato, Y., Suzuki, H., Yoshizato, T., Yoshida, K., *et al.* (2017). Gene expression and risk of leukemic transformation in myelodysplasia. *Blood* 130, 2642-2653.

Stein, E.M., DiNardo, C.D., Pollyea, D.A., Fathi, A.T., Roboz, G.J., Altman, J.K., Stone, R.M., DeAngelo, D.J., Levine, R.L., Flinn, I.W., *et al.* (2017). Enasidenib in mutant-IDH2 relapsed or refractory acute myeloid leukemia. *Blood*.

Wang, F., Travins, J., DeLaBarre, B., Penard-Lacronique, V., Schalm, S., Hansen, E., Straley, K., Kernytsky, A., Liu, W., Gliser, C., *et al.* (2013). Targeted inhibition of mutant IDH2 in leukemia cells induces cellular differentiation. *Science* 340, 622-626.

Yen, K., Travins, J., Wang, F., David, M.D., Artin, E., Straley, K., Padyana, A., Gross, S., DeLaBarre, B., Tobin, E., *et al.* (2017). AG-221, a First-in-Class Therapy Targeting Acute Myeloid Leukemia Harboring Oncogenic IDH2 Mutations. *Cancer Discovery*.

REVIEWER COMMENTS

Reviewer #1 (Remarks to the Author):

In this revised paper, “Evolution of AML genome and epigenome with IDH inhibitors and their association with clinical response and resistance,” Wang, Morita, DiNarto, and colleagues updated the analysis of their profiling data (DNA sequencing, RNA sequencing and cytosine methylation profiling) in longitudinally collected specimens from 68 IDH1/IDH2-mutant AML patients treated with IDH inhibitors (IDHi). Overall the edits have been responsive and I only have a few comments left:

1) They specifically noted that their cohort might have been underpowered to detect molecular predictors for primary resistance to IDH inhibitor, consistent with their data that the RUNX1 mutation was the only statistically significant marker associated with primary resistance. They have acknowledged this limitation and added the statement in the discussion, as well as a new supplementary table.

2) They mention the previous Harding et al paper on isoform switching, but I would still not recommend this term, since it normally refers to splicing isoforms, NOT switching to a different gene (homolog). That would be homolog switching, not isoform. The field uses more commonly “gene switching” or “resistance gene shift,” but not isoform switching. It’s OK to mention that some have called it isoform switching, but it is a really, really bad use of the term that only invites confusion. A simple google search will also reveal this. See here for the more canonical and appropriate use of isoform switching:

<https://mcr.aacrjournals.org/content/15/9/1206>

3) They note they did not see a CH difference between DTA+ and DTA- cases, and this would be good to mention in the discussion.

4) The methods are updated and I was able successfully to access their GEO submission (GSE153349).

5) The discussion could use an exploration of combined epigenetic therapies, which may help with this clinical resistance issue (see for example here):

<https://cancerdiscovery.aacrjournals.org/content/7/5/494>

Thank you.

Reviewer #2 (Remarks to the Author):

No further comments

We would like to again thank the reviewers for their effort in reviewing our revised manuscript. We are most grateful for the comments which have tremendously helped us to improve considerably the manuscript. Our point by point response is outlined below. We thank you in advance for your attention and hope you will find our manuscript suitable for publication in *Nature Communications*.

Reviewer #1:

1) They specifically noted that their cohort might have been underpowered to detect molecular predictors for primary resistance to IDH inhibitor, consistent with their data that the RUNX1 mutation was the only statistically significant marker associated with primary resistance. They have acknowledged this limitation and added the statement in the discussion, as well as a new supplementary table.

RE: We thank the reviewer for the acknowledgment.

2) They mention the previous Harding et al paper on isoform switching, but I would still not recommend this term, since it normally refers to splicing isoforms, NOT switching to a different gene (homolog). That would be homolog switching, not isoform. The field uses more commonly “gene switching” or “resistance gene shift,” but not isoform switching. It’s OK to mention that some have called it isoform switching, but it is a really, really bad use of the term that only invites confusion. A simple google search will also reveal this. See here for the more canonical and appropriate use of isoform switching: <https://mcr.aacrjournals.org/content/15/9/1206>

RE: We are in total agreement with the reviewer on the confusing nature of “isoform switching” term. In the revised manuscript, we used “homolog switch” to avoid the confusion. Thank you for this remark.

3) They note they did not see a CH difference between DTA+ and DTA- cases, and this would be good to mention in the discussion.

RE: We have added our observation in the revised manuscript (page 6).

4) The methods are updated and I was able successfully to access their GEO submission (GSE153349).

RE: We thank the reviewer for the acknowledgment.

5) The discussion could use an exploration of combined epigenetic therapies, which may help with this clinical resistance issue (see for example here): <https://cancerdiscovery.aacrjournals.org/content/7/5/494>

RE: We think this reference is helpful in discussing novel combination strategies to overcome resistance and added the reference in the revised paper (ref: 26).

Reviewer #2:

No further comments.

RE: We thank the reviewer for the review of our revised paper.

REVIEWERS' COMMENTS

Reviewer #2 (Remarks to the Author):

No further comments.